# NAK-associated protein 1/NAP1 activates TBK1 to ensure accurate mitosis and cytokinesis

Swagatika Paul[1], Shireen A. Sarraf[2], Ki Hong Nam[3], Leila Zavar[4], Nicole DeFoor[4], Sahitya Ranjan Biswas[5], Lauren E. Fritsch[5], Tomer M. Yaron[6,7], Jared L. Johnson[6], Emily M. Huntsman[6,7], Lewis C. Cantley[6,8,9], Alban Ordureau[3], and Alicia M. Pickrell[4]

Subcellular location and activation of Tank Binding Kinase 1 (TBK1) govern precise progression through mitosis. Either loss of activated TBK1 or its sequestration from the centrosomes causes errors in mitosis and growth defects. Yet, what regulates its recruitment and activation on the centrosomes is unknown. We identified that NAK-associated protein 1 (NAP1) is essential for mitosis, binding to and activating TBK1, which both localize to centrosomes. Loss of NAP1 causes several mitotic and cytokinetic defects due to inactivation of TBK1. Our quantitative phosphoproteomics identified numerous TBK1 substrates that are not only confined to the centrosomes but are also associated with microtubules. Substrate motifs analysis indicates that TBK1 acts upstream of other essential cell cycle kinases like Aurora and PAK kinases. We also identified NAP1 as a TBK1 substrate phosphorylating NAP1 at S318 to promote its degradation by the ubiquitin proteasomal system. These data uncover an important distinct function for the NAP1–TBK1 complex during cell division.

## Introduction

Successful cell division is dependent on the precise and timely transition between different cell cycle phases, which is regulated by dynamic changes in protein phosphorylation. Thus, protein kinases play a vital role in orchestrating almost every step of cell division such as centrosome maturation, chromatin condensation, spindle assembly formation, sister chromatid segregation, and cytokinesis (Enserink and Kolodner, 2010; Nasa and Kettenbach, 2018; Nigg, 2001; Seki et al., 2008). Entry into mitosis is marked by the highest incidences of protein phosphorylation and kinase activity (Dephoure et al., 2008; Olsen et al., 2010). Therefore, impaired or aberrant kinase activity often leads to errors in all these cell cycle events, which consequently become the underlying cause of developmental defects (Colas, 2020; Schneider and Ellenberg, 2019) or abnormal cell proliferation leading to cancer (Huang et al., 2021; Singh et al., 2017).

Tank Binding Kinase 1 (TBK1) is one such kinase, which is known to be activated on the centrosomes during mitosis (Pillai et al., 2015; Sarraf et al., 2019) and is also often overexpressed in certain cancer types (Chen et al., 2017; Uhlen et al., 2017; Wei et al., 2014). Genetic loss of TBK1 leads to embryonic lethality in mice (Bonnard et al., 2000), and its loss results in mitotic defects in cancer cell lines (Maan et al., 2021; Pillai et al., 2015; Sarraf

et al., 2019). Interestingly, the sequestration of activated TBK1 away from the centrosomes also disrupts mitosis in neural epithelial stem cells and radial glia during Zika virus infection (Onorati et al., 2016). Our previous work has shown that sequestration of active TBK1 to the mitochondria during mitophagy, the selective degradation of mitochondria, also blocks mitosis because TBK1 can no longer localize on the centrosomes (Sarraf et al., 2019). Thus, both proper activation and subcellular localization of TBK1 are essential for mitotic progression. Yet, the upstream regulation of TBK1 during mitosis is unknown, and we do not completely understand the function of activated TBK1 on the centrosomes.

Activation of TBK1 depends on its binding to an adaptor protein, which induces a conformational change leading to trans-autophosphorylation on serine 172 of the kinase domain of TBK1 (Fu et al., 2018; Larabi et al., 2013; Li et al., 2016; Zhang et al., 2019). Interaction with the adaptor protein not only activates the TBK1 kinase domain but may drive its subcellular localization to different organelles to regulate distinct signaling pathways (Goncalves et al., 2011; Heo et al., 2015; Thurston et al., 2016). From extensive studies examining its regulation during innate immune signaling, autophagy, and mitophagy, we know

[1]Graduate Program in Biomedical and Veterinary Sciences, Virginia-Maryland College of Veterinary Medicine, Blacksburg, VA, USA; [2]Biochemistry Section, National Institutes of Neurological Disorders and Stroke, National Institutes of Health, Bethesda, MD, USA; [3]Cell Biology Program, Sloan Kettering Institute, Memorial Sloan Kettering Cancer Center, New York, NY, USA; [4]School of Neuroscience, Virginia Polytechnic Institute and State University, Blacksburg, VA, USA; [5]Translational Biology, Medicine, and Health Graduate Program, Virginia Polytechnic Institute and State University, Roanoke, VA, USA; [6]Meyer Cancer Center, Weill Cornell Medicine, New York, NY, USA; [7]Englander Institute for Precision Medicine, Institute for Computational Biomedicine, Weill Cornell Medicine, New York, NY, USA; [8]Department of Cell Biology, Harvard Medical School, Boston, MA, USA; [9]Dana-Farber Cancer Institute, Harvard Medical School, Boston, MA, USA.

Correspondence to Alicia M. Pickrell: alicia.pickrell@vt.edu.

that TBK1 has multiple binding partners/adaptors for each of these cellular processes. While TANK, SINTBAD, NAP1/AZI2, optineurin (OPTN), and STING (Bakshi et al., 2017; Clark et al., 2011b; Fujita et al., 2003; Gatot et al., 2007; Gleason et al., 2011; Goncalves et al., 2011; Ryzhakov and Randow, 2007; Tanaka and Chen, 2012; Zhang et al., 2019) are the major TBK1 adaptors during innate immune signaling, OPTN, TAX1BP1, and NDP52 (Heo et al., 2015; Lazarou et al., 2015; Moore and Holzbaur, 2016; Richter et al., 2016) can activate TBK1 during mitophagy. However, the adaptor or adaptors required for TBK1 activation and recruitment during mitosis is unknown.

Along with activation, localization of TBK1 on the centrosomes is also essential for mitosis (Onorati et al., 2016; Sarraf et al., 2019). This localization brings activated TBK1 in proximity to microtubules as centrosomes are the microtubule organizing centers. Past studies have identified a few of the TBK1 substrates on the centrosomes (Kim et al., 2013; Maan et al., 2021; Pillai et al., 2015). Whether TBK1 functions only to phosphorylate centrosomal proteins during mitosis or also regulates microtubule-binding proteins remains to be determined. The complete landscape of the proteins targeted by TBK1 during mitosis remains unclear. Identifying all of the mitotic substrates would offer mechanistic insight into pathways modulated by TBK1 to ensure proper chromosomal segregation.

We show that NAP1/AZI2, whose function has only been described in innate immunity to trigger Type I interferon or NF-κB signaling (Fujita et al., 2003; Sasai et al., 2005, 2006), is necessary for TBK1 activation during mitosis. We discovered NAP1 to be a centrosomal protein that regulates proper cell division by binding and activating TBK1 during mitosis. Loss of either NAP1 or TBK1 results in the accumulation of binucleated and multinucleated cells due to the several mitotic and cytokinetic defects observed across several cell lines. We also describe a new function for both proteins, as our data suggests that they are also implicated in cytokinesis. Interestingly, NAP1 levels during mitosis are tightly regulated by TBK1. Activated TBK1 phosphorylates NAP1 on serine 318, flagging it for ubiquitin proteasomal degradation (UPS). Through unbiased quantitative phosphoproteomics analysis during mitosis, we also uncovered unidentified TBK1 substrates, which implicate its upstream effects on other cell cycle kinases such as Aurora A and Aurora B. This NAP1–TBK1 signaling axis during mitosis is distinct in its function during innate immunity.

## Results

### NAP1/AZI2 is required for TBK1 activation during mitosis
Activation of TBK1 is reliant upon its binding to adaptor proteins, which initiates higher-order oligomerization of the TBK1-adaptor complex, leading to trans-autophosphorylation at serine 172 (p-TBK1; Fu et al., 2018; Larabi et al., 2013; Ma et al., 2012). Previous identification of these adaptor proteins for TBK1 activation in other cellular contexts displayed overlap and redundancy. For example, during mitophagy, TAX1BP1, optineurin, and NDP52 have been found to be necessary for TBK1 activation (Heo et al., 2015; Lazarou et al., 2015; Moore and Holzbaur, 2016; Richter et al., 2016). Optineurin and NDP52 are also the bound

adaptor for TBK1 during certain innate immune stimuli (Fu et al., 2018; Li et al., 2016; Morton et al., 2008; Thurston et al., 2009). Despite these observations, the adaptor or adaptors required for TBK1 during mitosis is unknown. Therefore, we screened known TBK1 adaptors to determine whether any of these proteins were responsible for its activation during mitosis.

First, we investigated TBK1 adaptors restricted to innate immune signaling, TANK, SINTBAD, and NAP1, by generating cell lines in which these proteins were stably knocked down (KD) in HeLa cells (Fig. 1, A–C). Reductions in TANK and SINTBAD did not alter TBK1 activation during mitosis (Fig. 1, D–G). However, loss of SINTBAD did alter TBK1 activation in asynchronous cells (Fig. 1, F and H; see Discussion). NAP1 KD cells displayed a reduction of p-TBK1 during mitosis, indicating NAP1 could be required for TBK1 activation (Fig. 1, I and J). We also assessed p-TBK1 levels in a cell line lacking five autophagy-related adaptors that TBK1 either binds or phosphorylates: NBR1, TAX1BP1, OPTN, p62, and NDP52 as these proteins have been implicated in TBK1 activation during mitophagy, xenophagy, and innate immunity (Heo et al., 2015; Lazarou et al., 2015; Moore and Holzbaur, 2016; Pourcelot et al., 2016; Ravenhill et al., 2019; Richter et al., 2016; Thurston et al., 2009; Vargas et al., 2019; Wild et al., 2011; Wong and Holzbaur, 2014), but found no difference between p-TBK1 levels when cells were synchronized (Fig. 1, K and L).

To confirm that TBK1 activation during mitosis was dependent on NAP1, we generated two independent NAP1 CRISPR knockout (KO) clones targeting exon 4 (Fig. 2 A). Both NAP1 KO HeLa clones displayed decreased p-TBK1 levels during mitosis (Fig. 2, A and B). This result was specific to NAP1 because p-TBK1 levels were restored during mitosis upon stable reintroduction of the protein (Fig. 2, C and D).

### NAP1 KO cells have mitotic and cytokinetic defects like those lacking TBK1
Previously, we and others have shown that loss of TBK1 led to slowed cell growth, decreased number of mitotic cells in asynchronous conditions, and an increased prevalence of multinucleated cells (Kim et al., 2013; Onorati et al., 2016; Pillai et al., 2015; Sarraf et al., 2019). However, a more thorough analysis of the defects in cell division that led to these observations has not been performed. Therefore, we characterized cell division defects in NAP1 and TBK1 KO HeLa cells to compare whether these lines phenocopied each other. In asynchronous conditions, NAP1 KO cells exhibited slower growth rates, fewer number of mitotic cells, an increase in the number of bi- and multinucleated cells, and an increased number of cells displaying abnormal mitotic division (Fig. 2, E–I). We further characterized the types of abnormal mitotic defects between these KO lines. Both genotypes had a significant prevalence of monopolar spindles and splayed/unfocused spindle poles, but TBK1 KO had a significantly higher percentage of lagging chromosomes/acentric fragments while NAP1 KO cells had a higher percentage of cells with multipolar spindles (Fig. 2, J and K).

To further characterize how NAP1 and TBK1 regulate the progression of cell division, we also evaluated if cytokinetic defects were present. Both KO lines had a significantly higher percentage of cytokinetic cells in asynchronous conditions, and

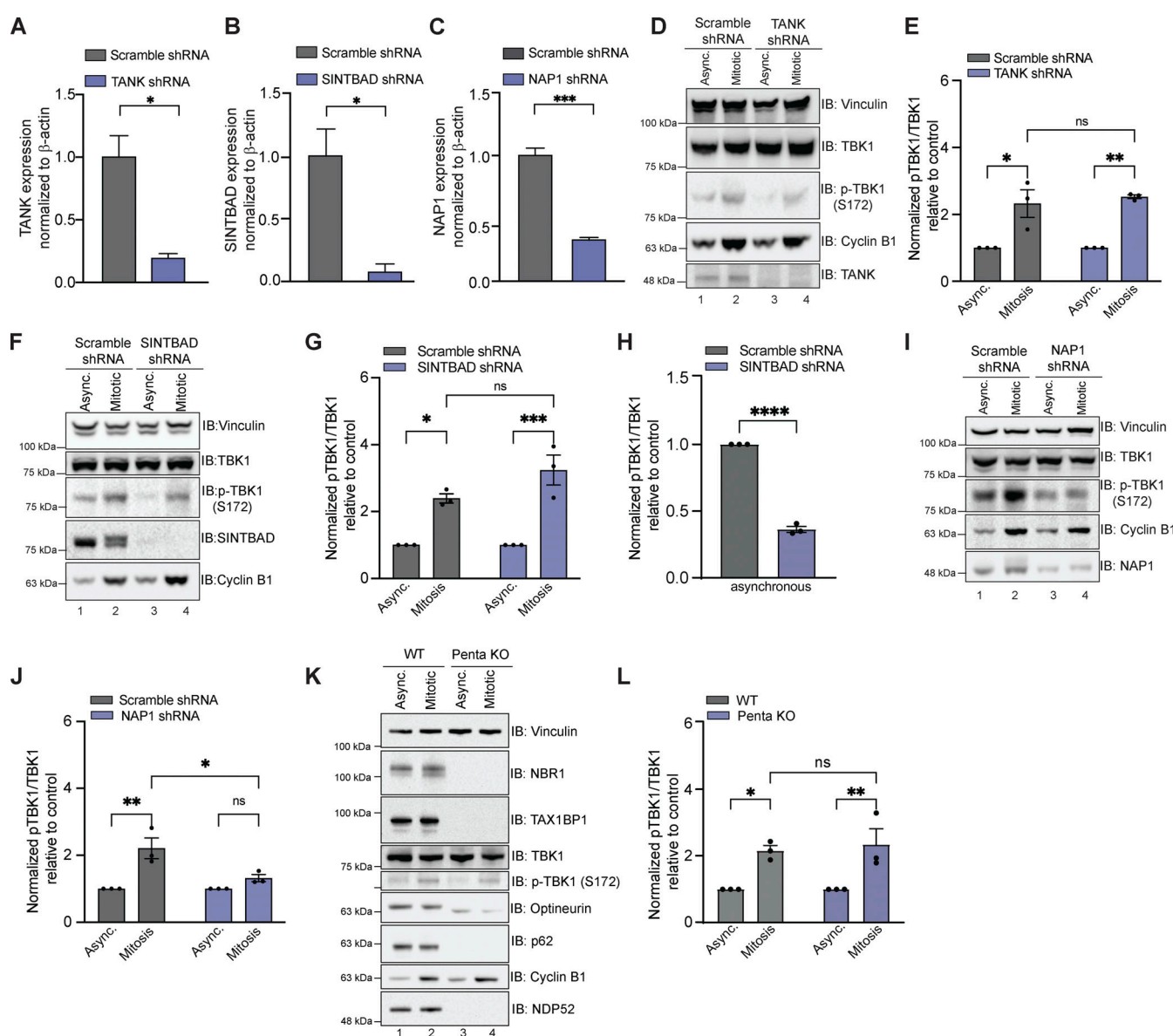

**Figure 1. Other known TBK1 adaptors except NAP1 are not required for TBK1 activation during mitosis. (A–C)** qRT-PCR showing relative expression of TANK1 (A), SINTBAD (B), and NAP1 (C) mRNA levels normalized to β-actin in HeLa cells stably expressing shRNAs. $n = 3$ independent experiments. Error bars ± SD. **(D–J)** Representative Western blots and semiquantitative analysis of p-TBK1/TBK1 levels normalized to vinculin in asynchronous and synchronized mitotic cells from scramble and cell lines stably expressing TANK (D and E), SINTBAD (F and G), and NAP1 (I and J) shRNA, respectively. Semi-quantitative analysis of p-TBK1/TBK1 levels normalized to vinculin in asynchronous SINTBAD (H) shRNA expressing cells. Nocodazole was used for synchronization. $n = 3$ independent experiments. Error bars ± SEM. **(K and L)** Representative Western blot and semiquantitative analysis of p-TBK1/TBK1 levels normalized to vinculin during mitosis in WT and Penta KO HeLa cells lacking NBR1, TAX1BP1, optineurin, NDP52, and p62 in asynchronous and mitotic cells. RO-3306 was used for synchronization. $n = 3$ independent experiments. Error bars ± SEM. One dot = one independent experiment. Unpaired Student's $t$ test or one-way ANOVA was performed for all statistical analysis. *$P < 0.05$, **$P < 0.01$, ***$P < 0.001$, ns = not significant. Source data are available for this figure: SourceData F1.

these cells displayed a significantly higher number of cytokinetic defects (Fig. 2, L–N). Both KO lines exhibited unequal cytokinesis, as well as multipolar cytokinesis at a higher percentage than the parental line (Fig. 2, N and O).

We attempted to use the near diploid RPE-1 and DLD-1 cell lines to confirm our findings in HeLa cells but were unable to generate either knockout or stable shRNA-mediated knockdown of NAP1 in both cell lines due to excessive cell death. Using transient viral transduction of NAP1 shRNA over 36 h, we generated a reduction of NAP1 (Fig. S1 A). NAP1 KD in DLD-1 cells

had reduced p-TBK1 levels during mitosis (Fig. S1 A). From the mitotic analysis on transient NAP1 KD DLD-1 cells, we observed a decrease in the mitotic index and an increased percentage of bi- and multinucleated cells (Fig. S1, B–E). However, there was an insufficient number of mitotic cells found to perform in-depth mitotic defect analysis (Fig. S1 F). The high number of binucleated cells (Fig. S1 C), along with the significantly skewed frequency distribution compared with scramble control (Fig. S1 F) suggested cells underwent cytokinetic failure over the 36-h time period.

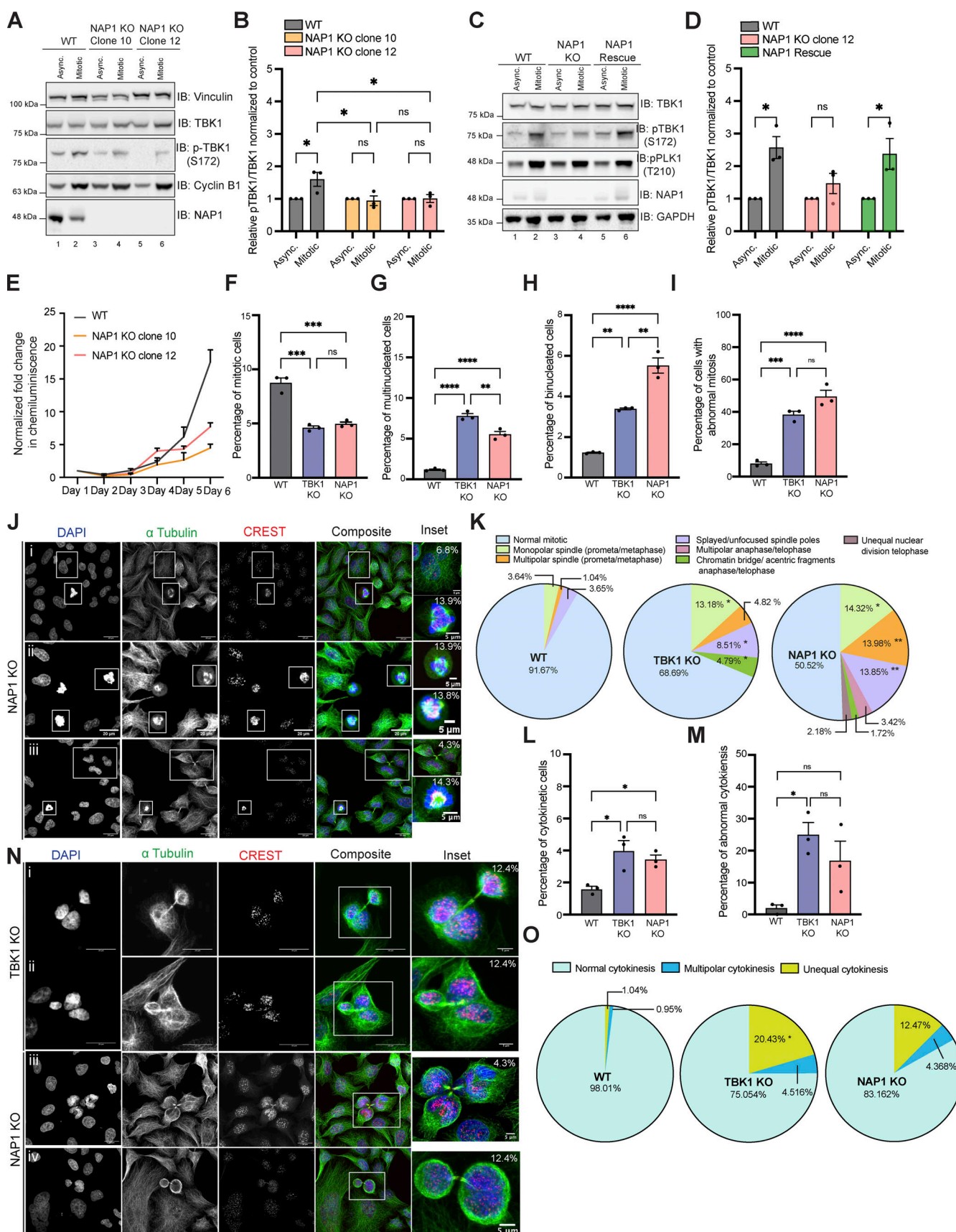

Figure 2. **NAP1 KO cells have mitotic defects similar to those lacking TBK1. (A and B)** Representative Western blot and semiquantitative analysis of p-TBK1/TBK1 levels normalized to vinculin in asynchronous and mitotic cells from WT HeLa, NAP1 KO clone 10, and NAP1 KO clone 12. Nocodazole was used

for synchronization. *n* = 3 independent experiments. Error bars ± SEM. **(C and D)** Representative Western blot and semiquantitative analysis of p-TBK1/TBK1 levels normalized to vinculin in asynchronous and synchronized mitotic cells from HeLa, NAP1 KO clone 12, and the stable NAP1 rescue line. RO-3306 was used for synchronization. *n* = 3 independent experiments. Error bars ± SEM. **(E)** Growth curve with normalized luminescence for WT HeLa, NAP1 KO clone 10, and NAP1 KO clone 12. Error bars indicate ± SEM. *n* = 2 experimental replicates. **(F–I)** Percentage of mitotic (F), multinucleated (G), binucleated (H), and abnormal mitotic (I) cells from an asynchronous population of WT HeLa, TBK1 KO, and NAP1 KO cells. Error bars indicate ± SEM; *n* = 3 independent experiments. For mitotic index, multinucleated and binucleated cell counts, and random fields of view were captured sampling ~1,000 cells per biological replicate from each genotype. For abnormal mitotic cell counts, random fields of view were captured to sample ~50 mitotic cells per biological replicate. *n* = 3 independent experiments from each genotype. **(J)** Representative confocal images of defects in NAP1 KO cells: insets (i) binucleated cell (top), multipolar metaphase (bottom); (ii) multipolar prometaphase (top), splayed/unfocused spindle (bottom); (iii) multipolar cytokinesis (top), monopolar prometaphase/metaphase (bottom). DAPI (blue) was used as a nuclear counterstain, α-tubulin for cytoskeleton staining (green), and CREST for kinetochore staining (red). Scale bar, 20 μm, insets, 5 μm. Percentages in the upper right corner of the insets display the percentage of that type of defect. **(K)** Pie charts representing the percentage of different types of mitotic defects found in WT HeLa, TBK1 KO, and NAP1 KO cells. At least 50 mitotic cells per biological replicate. *n* = 3 independent experiments from each genotype were analyzed. **(L and M)** Percentage of cytokinetic (L) and abnormal cytokinetic (M) cells from an asynchronous population of WT HeLa, TBK1 KO, and NAP1 KO. Error bars indicate ± SEM. *n* = 3 independent experiments. Random fields of view were captured sampling ~1,000 cells per biological replicate from each genotype. **(N)** Representative confocal images of cytokinetic defects seen in TBK1 KO (i and ii) and NAP1 KO (iii and iv): (i) unequal cytokinesis, (ii) multipolar cytokinesis, (iii) multipolar cytokinesis, (iv) unequal cytokinesis. DAPI (blue) was used as a nuclear counterstain, α-tubulin for cytoskeleton staining (green), and CREST for kinetochore staining (red). Scale bar, 20 μm, insets, 5 μm. Percentages in the upper right corner of the insets display the percentage of that type of mitotic defect. **(O)** Pie chart representing the percentage of different types of cytokinetic defects found in HeLa, TBK1, and NAP1 KO cells. Random fields of view were captured sampling ~30–40 cytokinetic cells per biological replicate from each genotype. *n* = 3 independent experiments. *P < 0.05 compared to HeLa. One dot = one independent experiment. One-way ANOVA was performed for all statistical analysis. *P < 0.05, **P < 0.01, ***P < 0.001, ****P < 0.0001, ns = not significant. Source data are available for this figure: SourceData F2.

Considering that the KD DLD-1 cells had NAP1 disrupted over a span of two cell divisions, we utilized the degradation tag system (dTAG; Nabet et al., 2018, 2020) for target-specific protein degradation to generate a FKBP$^{F36V}$-NAP1 DLD-1 cell line to allow for immediate and selective manipulation of NAP1 instead of relying on the temporal time scale required for KD efficiency to occur with viral transduction. This would allow us to characterize mitotic defects caused due to NAP1 loss within one mitotic division period and minimize cell death. The FKBP$^{F36V}$ variant allows for selective recognition by a dTAG ligand, like dTAG-13 or dTAG$^{V}$-1, to induce dimerization of the FKBP$^{F36V}$ fused NAP1 to the CRBN or VHL E3 ligase for ubiquitin proteosome degradation (UPS; Nabet et al., 2018, 2020; Fig. 3 A). After the addition of dTAG$^{V}$-1, NAP1 degradation occurred in both asynchronous and synchronized mitotic cells after 2 h, and p-TBK1 activation was dampened as compared with vehicle-treated mitotic lysates (Fig. 3, B–D).

To observe the consequences of NAP1 loss in DLD-1 cells over one cell division period in an asynchronous population, we treated the FKBP$^{F36V}$-NAP1 cell line for 20 h (Fig. 3, E and F). In line with our results when NAP1 was knocked down for 36 h, we observed a decline in mitotic index and an accumulation of binucleated cells, but multinucleated cells did not differ (Fig. 3, G–I). The lack of multinucleated cells was not completely unexpected as only one round of cell division occurred. The stress of the sudden degradation of NAP1 did lead to an increase in the number of dead cells after dTAG$^{V}$-1 treatment (Fig. 3 J). dTAG$^{V}$-1-treated FKBP$^{F36V}$-NAP1 cells had a significantly higher percentage of abnormal mitotic cells (Fig. 3 K) with many more mitotic defects (Fig. 3, L and M) than seen in the NAP1 KO HeLa line (Fig. 2, J and K). This result extended into cytokinesis where dTAG$^{V}$-1-treated FKBP$^{F36V}$-NAP1 cells had a higher percentage of cytokinetic cells that were abnormal with a significant number of unequal and multipolar cytokinetic cells (Fig. 3, N–Q). These data suggest that NAP1 is required for both mitosis and cytokinesis most likely through the activation of TBK1.

To understand how mitotic errors were generated, we performed live imaging on H2B–mCherry-expressing dTAG$^{V}$-1-treated FKBP$^{F36V}$-NAP1 cells 1 h prior to imaging or with a vehicle in asynchronous conditions (Video 1 and Fig. 3 R). Timing each stage of mitosis, we found a significant lag in prophase and prometaphase, while timing did not appear to differ during metaphase when compared with untreated cells (Fig. 3, R–U). A significant difference was also detected for the duration between the entry into mitosis and the onset of anaphase (Fig. 3 V). Watching the fate of these dTAG$^{V}$-1 treated cells, a significant number of cells did not complete mitosis normally, tried to compensate for these errors, or died (Fig. 3, W and X; and Videos 2, 3, 4, and 5) accounting for the increase in time during the early mitotic stages. This data also explains the accumulation of cells detected in prophase and prometaphase with NAP1 KD after 36 h of shRNA expression (Fig. S1 F). We conclude that NAP1 is required for accurate mitotic cell division.

## TBK1 selectively interacts with and binds NAP1 during mitosis

Next, we wanted to determine if there was an interaction between NAP1 and TBK1 during mitosis as the literature suggests that adaptor binding must occur for TBK1 activation (Larabi et al., 2013; Ma et al., 2012). We performed coimmunoprecipitation (co-IP) experiments by transiently overexpressing either full-length GFP-NAP1 (N' EGFP FL NAP1) or GFP-NAP1 lacking the TBK1 binding domain (N' EGFP NAP1 Δ230–270; Fig. 4 A) in HEK293T cells (Ryzhakov and Randow, 2007) and also performed the reciprocal co-IP by stably expressing two different N'FLAG-HA-TBK1 rescue constructs (Fig. 4 B) in TBK1 KO HeLa cells as our previous data indicated that TBK1 levels are tightly regulated inside the cell (Sarraf et al., 2019). Phosphorylated endogenous TBK1 was enriched upon the immunoprecipitation of N' EGFP FL NAP1 with increased binding during mitosis (Fig. 4, C and D). The binding of p-TBK1 was abolished in both asynchronous and mitotic conditions when NAP1 lacked its TBK1 binding domain (Fig. 4 C). Endogenous NAP1 was enriched upon immunoprecipitation of full-length TBK1 (N'FLAG-HA FL TBK1)

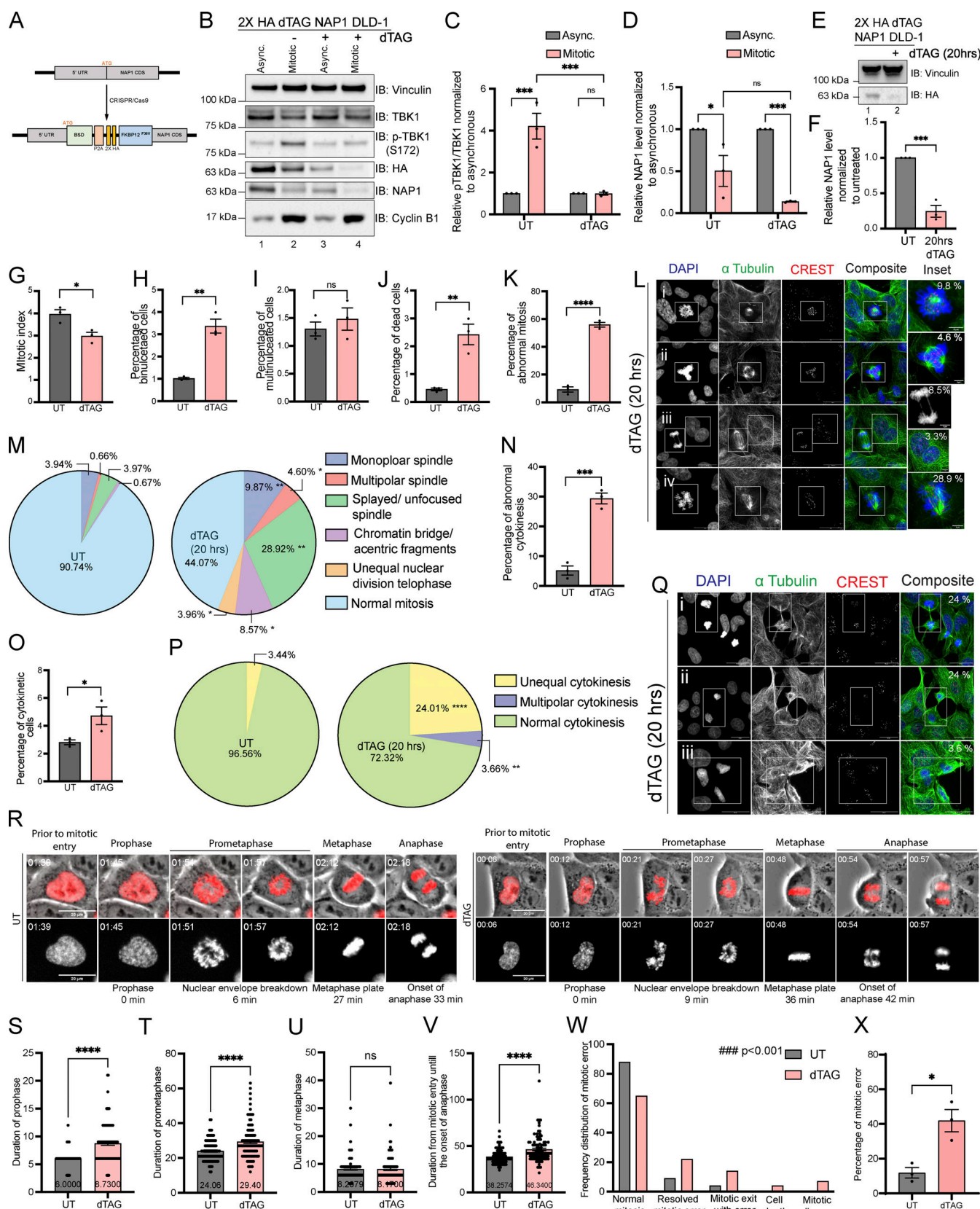

Figure 3. **NAP1 loss in a near diploid cell line causes mitotic and cytokinetic defects. (A)** Cartoon diagram of dTAG knock-in constructs designed to add FKBP12$^{F36V}$ to the N-terminus of NAP1. **(B–D)** Representative Western blot and semiquantitative analysis of p-TBK1/TBK1 (C) and NAP1 (D) levels normalized to vinculin in asynchronous and mitotic cells from FKBP12$^{F36V}$-NAP1 DLD-1 cells. RO-3306 was used for synchronization prior to mitotic release. Cells were treated with dTAG$^V$-1 for 3 h in asynchronous conditions and 3 h prior and during release in mitotic conditions. $n$ = 3 independent experiments. Error bars ±

SEM. **(E and F)** Representative Western blot and semiquantitative analysis of NAP1 levels normalized to vinculin in asynchronous cells with or without dTAG$^V$-1 treatment for 20 h. $n$ = 3 independent experiments. Error bars ± SEM. **(G–K)** Percentage of mitotic (G), binucleated (H), multinucleated (I), dead (J) and abnormal mitotic (K) cells from an asynchronous population of FKBP12$^{F36V}$-NAP1 DLD-1 cells untreated or treated with 20 h of dTAG$^V$-1. Error bars indicate ± SEM; $n$ = 3 independent experiments. For mitotic index, multinucleated, and binucleated cell counts, random fields of view were captured sampling ∼800–900 cells per biological replicate from each genotype. For abnormal mitotic cell counts, random fields of view were captured to sample ∼50 mitotic cells per biological replicate. $n$ = 3 independent experiments from each genotype. **(L)** Representative confocal images of the defects seen in FKBP12$^{F36V}$-NAP1 DLD-1 cells after 20 h of dTAG$^V$-1 treatment: (i) monopolar spindle, (ii) multipolar spindle, (iii) chromatin bridge/acentric fragment during anaphase (top) and binucleated cell (bottom), (iv) unfocused spindle poles. DAPI (blue) was used as a nuclear counterstain, α-tubulin for cytoskeleton staining (green), and CREST for kinetochore staining (red). Scale bar, 20 μm; insets, 5 μm. Percentages in the upper right corner of the insets display the percentage of that type of mitotic defect. **(M)** Pie charts representing the percentage of different types of mitotic defects found in untreated and dTAG$^V$-1 treated FKBP12$^{F36V}$-NAP1 DLD-1 cells for 20 h. At least 50 mitotic cells per biological replicate. $n$ = 3 independent experiments from each genotype were analyzed. **(N and O)** Percentage of cytokinetic (N) and abnormal cytokinetic (O) cells from untreated and dTAG$^V$-1 treated FKBP12$^{F36V}$-NAP1 cells for 20 h. Error bars indicate ± SEM. Random fields of view were captured sampling ∼800–900 cells per biological replicate. $n$ = 3 independent experiments from each genotype. **(P)** Pie chart representing the percentage of different types of cytokinetic defects found in untreated and dTAG$^V$-1 treated FKBP12$^{F36V}$-NAP1 DLD-1 cells for 20 h. Random fields of view were captured sampling ∼30–40 cytokinetic cells per biological replicate. $n$ = 3 independent experiments from each genotype. **(Q)** Representative confocal images of cytokinetic defects seen in FKBP12$^{F36V}$-NAP1 DLD-1 cells after 20 h of dTAG$^V$-1 treatment: (i and ii) unequal cytokinesis, (iii) multipolar cytokinesis. DAPI (blue) was used as a nuclear counterstain, α-tubulin for cytoskeleton staining (green), and CREST for kinetochore staining (red). Scale bar, 20 μm. Percentages in the upper right corner of the insets display the percentage of that type of mitotic defect. **(R)** Representative time-lapse images of the FKBP12$^{F36V}$-NAP1 DLD-1 cells from UT (left) and dTAG$^V$-1 treated (right). The actual experimental time from the beginning of the time-lapse imaging is displayed at the top left corner. The quantified duration of each early mitotic stage is displayed at the bottom. Scale bar, 20 μM. H2B–mCherry was used as the nuclear marker. **(S–V)** Duration of (S) prophase, (T) prometaphase, (U) metaphase, and (V) mitotic entry until the onset of anaphase in UT and dTAG$^V$-1 treated FKBP12$^{F36V}$-NAP1 DLD-1 cells from time-lapse imaging. Cells were pretreated with either DMSO for untreated or dTAG$^V$-1 treated group for 1 h prior to the imaging. The cells were followed up to 4–5 h. Random fields of view were imaged to capture at least 100 mitotic cells from three independent biological replicates for each group. Error bars indicate ± SEM. One dot = one cell. **(W)** Frequency distribution graph for different observed mitotic errors from the time-lapse imaging analysis sampled from ∼100 mitotic cells in UT and dTAG$^V$-1 treated FKBP12$^{F36V}$-NAP1 DLD-1 cells. Kolmogorov–Smirnov nonparametric test was used to analyze the differences between the frequency distribution. **(X)** Percentage of cells with a mitotic error in UT and dTAG$^V$-1 treated FKBP12$^{F36V}$-NAP1 DLD-1 cells from the time-lapse imaging. At least 100 cells from three independent biological replicates for each group were analyzed. Error bars indicate ± SEM. One dot = one experimental replicate for (C, D, F, G–K, N, O, and X). Student's $t$ test was performed for all statistical analyses. *$P < 0.05$, **$P < 0.01$, ***$P < 0.001$, ****$P < 0.0001$, ns = not significant. Source data are available for this figure: SourceData F3.

during mitosis (Fig. 4, E and F). This binding did not occur in rescue lines when the known C-terminal TBK1 adaptor binding site was deleted (N'FLAG-HA TBK1ΔC'; Fig. 4 G). Additionally, we verified the enriched binding between NAP1 and TBK1 during mitosis by performing immunoprecipitation of endogenous TBK1 in DLD-1 cells (Fig. 4, H and I).

Next, we sought to corroborate our IP data with experiments to detect the subcellular location of NAP1 during mitosis. Due to the unavailability of antibodies suitable to detect endogenous NAP1 by immunofluorescence, we transiently expressed N'EGFP NAP1 in HeLa cells. We found that NAP1 colocalized with p-TBK1 on the centrosomes of mitotic cells (Fig. 4, J and K). As NAP1 binding with TBK1 is necessary for TBK1 activation, NAP1 KO cells displayed a significant decrease in p-TBK1 intensity on the centrosomes compared with WT (Fig. 4, L and M). Interestingly, this decrease in signal was accompanied by a reduced area stained by p-TBK1 in the NAP1 KO cells (Fig. 4 N). To verify if adding NAP1 back to the cells can rescue this phenotype, we used FKBP$^{F36V}$-NAP1 cells to first deplete NAP1 with dTAG$^V$-1 treatment for 2 h and perform a 4-h washout to restore NAP1 levels (Fig. 4, O and P). We observed a significant decrease in p-TBK1 fluorescence intensity as well as area stained in the dTAG$^V$-1 treated cells, which were rescued after washout (Fig. 4, Q–S). These data suggest that activation of TBK1 during mitosis is dependent on NAP1 binding.

Although our data suggest that NAP1 has a separate function in regulating TBK1 during mitosis, previous studies of NAP1 have been limited to understanding its role during innate immunity (Fujita et al., 2003; Goncalves et al., 2011; Ryzhakov and Randow, 2007; Sasai et al., 2005). It is possible that the binding of NAP1 to TBK1 during the cell cycle could aberrantly activate innate immune pathways. NAP1 is an adaptor for TBK1 during innate immune response, in which TBK1 on the endoplasmic reticulum phosphorylates the transcription factor, interferon regulator factor 3 (IRF3), causing its translocation to the nucleus, which in turn results in the stimulation of the Type I interferon response (Fitzgerald et al., 2003; Goncalves et al., 2011; Ryzhakov and Randow, 2007; Sasai et al., 2005; Sharma et al., 2003).

Using a human monocytic cell line that is highly responsive to immunogenic stimuli, we first confirmed that this cell line displayed TBK1 activation during mitosis (Fig. S2 A) and after exposure to innate immune responsive stimuli (LPS and poly I:C; Fig. S2 B). We then assessed the expressions of a few selected cytokine genes, such as *Il10*, *Il6*, and *TNFA*, which are known to be upregulated upon LPS and poly I:C treatment (Fig. S2, C and D). During mitosis, the expression of these genes was either downregulated or only slightly upregulated, in the case of TNF-α, but not reaching the upregulation levels seen during immune stimuli (Fig. S2, C–E), indicating that TBK1 activation during mitosis most likely does not trigger a robust innate immune response. Phosphorylation of either IRF3 or an alternative innate immune kinase (IKKε) that works in conjunction with TBK1 and shares high sequence homology (Tojima et al., 2000) occurred only in response to immunogenic stimuli (LPS and poly I:C) and not during mitosis (Fig. S2 F). Together, these data suggest there exists a distinct mitotic NAP1–TBK1 signaling axis at centrosomes.

### NAP1 expression level is controlled by the UPS

We consistently observed a reduction or upper molecular weight smear in NAP1 protein during mitosis across four different cell lines (HeLa, DLD-1, THP1, and RPE-1, and an additional near-

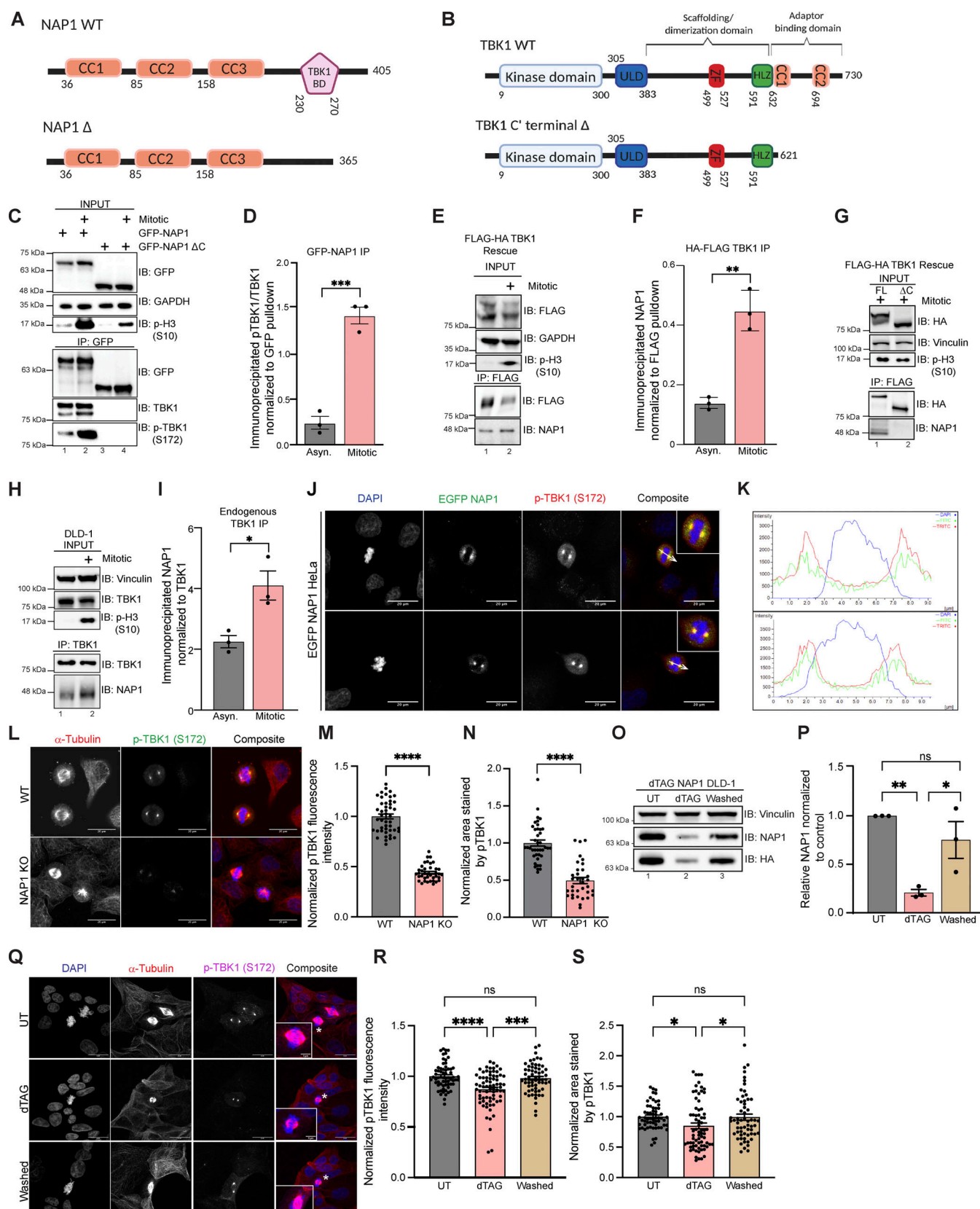

Figure 4. **NAP1 binds to TBK1 during mitosis and is localized to centrosomes. (A and B)** Cartoon (A) represents protein domains of full-length (FL) NAP1 and NAP1 lacking the TBK1 binding domain (Δ230–270). Cartoon (B) represents protein domains of full length (FL) TBK1 and TBK1 lacking adaptor binding domain (Δ C' terminal truncation). **(C)** Representative immunoblots of the pulldowns of transiently expressed GFP-NAP1 or GFP-NAP1 Δ230–270 (lacking TBK1 binding domain) in asynchronous and synchronized mitotic HEK293T cells. Nocodazole was used for cell synchronization. **(D)** Semiquantitative pTBK1/TBK1

levels after normalization to the pulldown efficiency of GFP-NAP1. Error bars indicate ± SEM; *n* = 3 independent experiments. **(E)** Representative immunoblots of the pulldown of HA-FLAG TBK1 in asynchronous and synchronized mitotic TBK1 rescue HeLa cells. Nocodazole was used for synchronization. **(F)** Semi-quantitative NAP1 levels after normalization to the pulldown efficiency of FLAG-TBK1. Error bars indicate ± SEM; *n* = 3 independent experiments. **(G)** Representative immunoblots of the pulldown of HA-FLAG TBK1 and HA-FLAG TBK1 Δ C′ (lacking adaptor binding domain) in synchronized mitotic TBK1 rescue HeLa cells. Nocodazole was used for synchronization. **(H)** Representative immunoblots of the pulldown of endogenous TBK1 in asynchronous and synchronized mitotic DLD-1 cells. Nocodazole was used for cell synchronization. **(I)** Semiquantitative endogenous NAP1 levels after normalization to the pulldown efficiency of TBK1. Nocodazole was used for cell synchronization. Error bars indicate ± SEM; *n* = 3 independent experiments. **(J)** Representative confocal images of HeLa cells transiently expressing EGFP-NAP1 with immunocytochemical detection of p-TBK1 (red). DAPI (blue) was used as a nuclear counterstain. White arrow depicts the area used for line scan analysis in H. Scale bar = 20 μm. **(K)** Line scan analysis of images in J generated from Nikon Elements software. **(L)** Representative confocal images of p-TBK1 expression on centrosomes from WT HeLa and NAP1 KO cells. DAPI (blue) was used as a nuclear counterstain (on composite image), α-tubulin for cytoskeleton staining (red), and p-TBK1 S172 conjugated 488 (green). Scale bar = 20 μm. **(M and N)** Relative intensity of p-TBK1 (M) and area stained by p-TBK1 (N) on the centrosomes of mitotic cells from WT HeLa and NAP1 KO cells. 40–50 mitotic cells per group were quantified from two biological replicates. Error bars indicate ± SEM. One dot = one cell. **(O and P)** Representative immunoblots (O) and semiquantitative NAP1 levels after normalization (P) of the from the lysates in UT, dTAG$^v$-1 treated, and washed FKBP12$^{F36V}$-NAP1 DLD-1 cells. 1 μM dTAG$^v$-1 was used for depleting the NAP1 level in the treated sample. For washed samples, NAP1 was first depleted with 2 h of dTAG$^v$-1 treatment and restored after washing and culturing in a normal medium for 4 h. Vinculin was used as a loading control. Error bars indicate ± SEM; *n* = 3 independent experiments. **(Q)** Representative confocal images of p-TBK1 expression on centrosomes from UT, dTAG$^v$-1 treated and washed FKBP12$^{F36V}$-NAP1 DLD-1 cells. DAPI (blue) was used as a nuclear counterstain, α-tubulin for cytoskeleton staining (red), and p-TBK1 S172 conjugated 647 (magenta). Scale bar = 20 μm, insets = 5 μm. **(R and S)** Relative intensity of pTBK1 (R) and area stained by p-TBK1 (S) on the centrosomes of mitotic cells from UT, dTAG$^v$-1 treated, and washed FKBP12$^{F36V}$-NAP1 DLD-1 cells. 60–70 mitotic cells per group were quantified from three biological replicates. Error bars indicate ± SEM. One dot = one independent experiment (D, F, I, and P). Unpaired Student's *t* test was performed for analysis between two groups, and one-way ANOVA was used for the analysis with three groups. *P < 0.05, **P < 0.01, ***P < 0.001, ****P < 0.0001 Source data are available for this figure: SourceData F4.

diploid cell line), independent of the method of synchronization (Figs. 1 I, 2, A and C, 3, B and D, S1 A, S2 A, and 5, A and B). This reduction in NAP1 during mitosis also occurred at the mRNA level (Fig. 5, C and D). However, this was not surprising as global transcription is repressed during mitosis (Prescott and Bender, 1962; Taylor, 1960). Inhibition of protein synthesis with cycloheximide over 6 h did not disrupt NAP1 protein levels in asynchronous conditions (Fig. 5, E–G). Also, the inhibition of lysosomal fusion with chloroquine, which blocks autophagy, showed no change to NAP1 levels in asynchronous conditions (Fig. 5, H–J). From these data, we conclude that NAP1 is a relatively stable protein except during mitosis.

To better define when and how NAP1 levels decreased during mitosis, we synchronized cells using the CDK1 inhibitor RO-3306 at G2 and released cells into mitosis. NAP1 was significantly reduced 10 min after release across cell lines (HeLa, RPE-1, and DLD-1), but the protein level mostly recovered during cytokinesis 60 min after release (Fig. 5, K–M, lanes 1–3). However, NAP1 levels did not degrade 10 min after release from G2 with the addition of the proteasome inhibitor, MG132 (Fig. 5, K–M, lanes 4, 5). This indicates that the UPS was involved in this reduction of NAP1 during mitosis.

To better characterize the exact time point at which NAP1 levels degrade during mitosis and return to levels seen during G1-asynchronous conditions, we used two approaches. Using p-PLK1 T210 and p-CDK1 Y15 as markers to indicate different stages during mitosis, we saw that NAP1 levels continuously degrade up until 40 min (metaphase/anaphase) after G2 release in RPE-1 cells (Fig. 5 N, lane 8). Levels began to increase and recover at the 50- and 60-min time points (cytokinesis), and the 90-min time point (asynchronous, G1; Fig. 5 N, lanes 9–11). To verify these results, we used the FKBP$^{F36V}$-NAP1 DLD-1 cell line which has endogenous NAP1 tagged with HA (Fig. 3 A). Using untreated cells, we performed HA immunofluorescence and found low NAP1 intensity in cells identified as being in prophase and prometaphase as compared with the nearby interphase cells,

while NAP1 expression appeared more robust during later stages of mitosis (Fig. 5 O). This data confirms that NAP1 is tightly regulated during mitosis.

### Quantitative phospho-proteomics pipeline identifies the mitotic downstream substrates of TBK1

The landscape of proteins targeted by TBK1 is still unclear, and the substrates that have been identified do not fully account for the various defects in mitosis and cytokinesis that occur with the loss of TBK1 or NAP1 (Figs. 2, 3, and S1; and Videos 1, 2, 3, 4, and 5; Kim et al., 2013; Maan et al., 2021; Pillai et al., 2015). Thus, we aimed to uncover additional mitotic targets of TBK1 through unbiased quantitative phosphoproteomic analysis. We synchronized two TBK1 KO HeLa cell lines, previously created in our work (Sarraf et al., 2019), with WT cells into mitosis. Furthermore, to account for potential off-target effects of CRISPR gene editing, we included experimental conditions in which TBK1 was blocked using a small-molecule inhibitor MRT67307 (Clark et al., 2011a; Fig. 6 A). While previous studies (Dephoure et al., 2008; Olsen et al., 2010) have detailed the global cell cycle and mitotic phosphorylation landscape, to our knowledge, an in-depth, focused quantitative analysis of TBK1-dependent changes in the phosphoproteome during mitosis has not been performed.

To broadly understand how the phosphoproteome changes during mitosis and identify potential TBK1 mitotic substrates, we used 16plex TMTpro to perform quantitative proteomics on synchronized and released cell extracts (Fig. 6 A). Tryptic peptides from whole-cell extracts were subjected to phosphopeptide enrichment, and samples were analyzed using 16plex TMTpro workflow (Li et al., 2020), with phosphopeptide intensities normalized with total protein abundance measured in parallel. In total, we quantified 8,650 proteins and 49,986 unique phosphopeptides mapped to 7,150 phosphoproteins from the phosphopeptide enrichment (Datas S1 and S2). Principal component analysis revealed reproducible replicate data, with 39.7% of the variance being driven by cell genotype and 27% driven by the

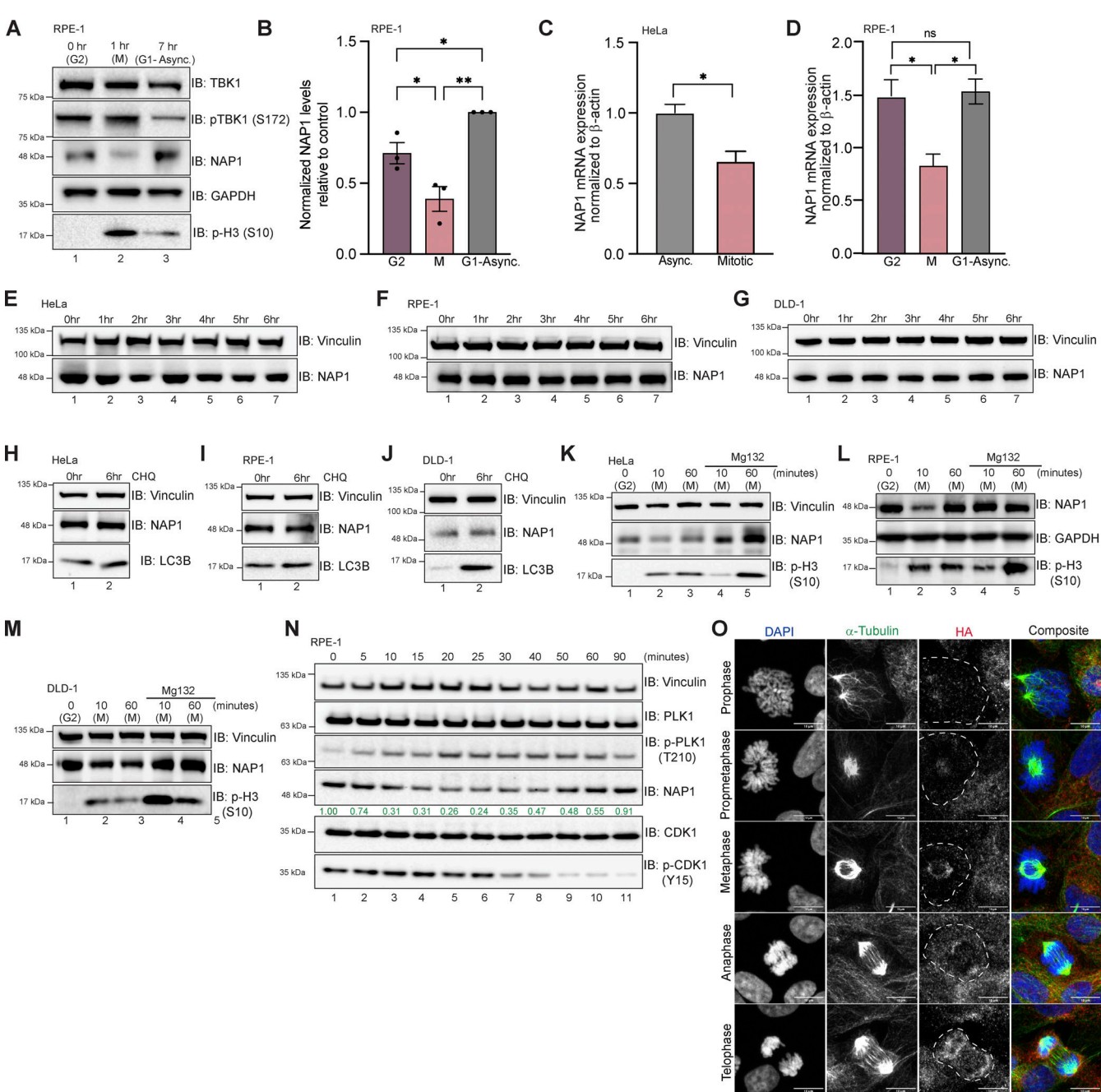

Figure 5. **NAP1 expression level is controlled by the UPS. (A and B)** Representative Western blot and semiquantitative analysis of NAP1 normalized to GAPDH in G2, mitotic (M), and G1-asynchronous conditions in RPE-1 cells. RO-3306 was used for synchronization. Error bars indicate ± SEM; $n$ = 3 independent experiments. **(C)** Relative NAP1 mRNA expression normalized β-actin in asynchronous and synchronized mitotic HeLa cells. Nocodazole was used for synchronization. Error bars indicate ± SD; $n$ = 3 independent experiments. **(D)** Relative NAP1 mRNA expression normalized to β-actin in G2, mitotic, and G1-asynchronous RPE-1 cells. Error bars indicate ± SD; $n$ = 3 independent experiments. Cells were synchronized at G2 using RO-3306. **(E–G)** Western blot analysis of NAP1 protein levels after cycloheximide treatment up to 6 h in HeLa (E), RPE-1 (F), and DLD-1 (G) cells. **(H–J)** Western blot analysis of NAP1 protein levels after 6 h of chloroquine treatment in HeLa (H), RPE-1 (I), and DLD-1 (J) cells. **(K–M)** Western blot analysis of NAP1 protein level after MG132 treatment followed by G2 release in HeLa (K), RPE-1 (L), and DLD-1 (M) cells. RO-3306 was used for synchronization. **(N)** Western blot analysis of NAP1 levels every 5 min after G2 release in RPE-1 cells. RO-3306 was used for synchronization. **(O)** Representative confocal images of FKBP12$^{F36V}$-NAP1 DLD-1 cells (untreated) with immunocytochemical detection of HA (red). DAPI (blue) was used as a nuclear counterstain and α-tubulin for cytoskeleton staining (green). A dashed outline indicates the cell border. Scale bar = 10 μm. One dot = one independent experiment. Student's $t$ test or one-way ANOVA was performed for all statistical analysis. *P < 0.05, **P < 0.01, ns = not significant. Source data are available for this figure: SourceData F5.

Figure 6. **Quantitative phosphoproteomics pipeline identifies the mitotic downstream substrates of TBK1. (A)** Workflow for TMTpro-based phosphoproteomics of ~20 M (million) mitotic cells. 16plex proteomics was performed in quadruplicate on samples harvested 1 h after mitotic release. **(B)** Volcano plots (log$_{10}$ [P value] versus log$_2$ ratio) for representing phosphorylation sites that are affected by the loss of TBK1. Proteins are shown in black circles (non-

significant) and red circles (Tier 1 significant). The circles were color-coded for the motif that most likely fit TBK1 (blue). Bolded black circles categorize known mitotic proteins (GO:0000278). The inset indicated additional color coding for the statistical analysis. **(C)** Volcano plot scoring each identified substrate against the 303 kinase motifs panel from Johnson et al. (2023) to determine statistically significant enriched activated kinases. **(D)** Motif analysis using the pLogo tool identifies the motif for the substrate candidates based on synthetic peptide analysis. The y-axis is the log odds of the binomial probability. **(E)** Gene Ontology terms enrichment analysis among the Biological Processes for enriched phosphosites Tier 2. Mitotic and cell cycle–related classes were significantly enriched (left panel). Enrichment map networks, each node represents a gene set (i.e., a GO term) and each edge represents the overlap between two gene sets (right panel). **(F and G)** Representative Western blot and semiquantitative analysis of pAurora A and B/Aurora A and B normalized to vinculin in asynchronous and mitotic HeLa and TBK1 KO cells. RO-3306 was used for synchronization. Error bars indicate ± SEM; n = 3 independent experiments. **(H and I)** Representative Western blot and semiquantitative analysis of pAurora A and B/Aurora A and B normalized to vinculin in asynchronous and mitotic FKBP12$^{F36V}$-NAP1 DLD-1 cells. dTAG$^v$-1 treated cells occurred 2 h prior to and during G2 release. RO-3306 was used for synchronization. Error bars indicate ± SEM; n = 3 independent experiments. One dot = one independent experiment. Student's t test was performed for WB statistical analysis. *P < 0.05, **P < 0.01, ns = not significant. Source data are available for this figure: SourceData F6.

---

small molecule inhibitor treatment (Fig. S3 A). From the ~50,000 unique phosphorylation sites quantified, we identified 660 sites in 493 proteins whose abundance was statistically decreased by greater than twofold (P < 0.01, 0.1% FDR) that differed between TBK1 genotypes (Fig. 6 B). Several highly significant phosphosites are previously reported bona fide substrates of TBK1 (e.g., SQSTM1, TAX1BP1), thus validating our experimental approach. Upon further analysis, we found several enriched sites associated with the gene ontology term for the mitotic cell cycle. This led us to consider them strong TBK1 substrate candidates.

Since the disturbance of a single kinase can affect numerous other signaling components, including other kinases, it is challenging to attribute changes observed in phosphoproteomic data to a specific kinase. Although such data provides an informative snapshot overview, it may only sometimes be easily traceable to a particular kinase. To gain more insight into our TBK1-focused dataset, we compared our phosphoproteomic data and the motifs surrounding our identified substrates with the kinase library data generated by the Cantley laboratory. This kinase library is based on 303 S/T kinases screened against a synthetic peptide library to define their individual motifs (Johnson et al., 2023). This allowed us to quantitively identify if other kinases were potentially affected by the loss of TBK1 and also sort out which potential substrates were directly TBK1-dependent. Using this methodology, we conducted a motif analysis to identify any sequence-level enrichment for sites statistically changed by greater than twofold (P < 0.05, 1% FDR). Our findings revealed that the substrate list predominantly consisted of Aurora, PAK, and TBK1/IKKε kinase motifs (Fig. 6, C and D; and Fig. S3 B). Of these, ~12.12% of sites were predicted to be putative TBK1/IKKε kinase substrates (>95th percentile; Fig. 6 B), and constitutive loss of TBK1 did not significantly impact the expression of many of the proteins detected (~1%; Fig. S4 A). Analyzing all ~50,000 sites with the kinase library, we found that substrates categorized as Aurora, PAK, and TBK1/IKKε kinase substrates (>95th percentile) were all significantly downregulated in TBK1 KO when compared with the parental TBK1$^{+/+}$ cells (Fig. S4 B).

Since CRISPR editing and small molecule inhibitors can independently cause off-target effects (Clark et al., 2012), we also cross-compared the four different experimental conditions. Upon examining all phosphosites, it was observed that substrates scored by Aurora, PAK, and TBK1/IKKε substrates did display potential off-target effects due to pharmacological

inhibition. However, upon further analysis of the data while considering genotype, it became apparent that a majority of Aurora and PAK kinase putative substrates are still down-regulated only upon TBK1 CRISPR deletion and unaffected by additional MRT67307 treatment (Fig. S4 C). After considering genotype and inhibitor treatment, we identified 63 sites in 56 proteins whose abundance was statistically increased by greater than twofold (P < 0.01, 1% FDR; Fig. S4 D). Despite this, many mitotic proteins, such as CEP97, NUSAP1, NES, etc., remain dysregulated in both analyses, indicating the impact of TBK1 on Aurora and PAK kinases (Fig. 6 B, Fig. S4 D, and Fig. S5 A; and Data S1).

To further evaluate the global effect of impairing TBK1's activity upon mitosis, we performed functional enrichment analysis based on Gene Ontology (GO) annotations for biological processes (Fig. 6 E and Fig. S5 B), molecular function (Fig. S5 C), and cellular component terms (Fig. S5 D). This analysis revealed a significant enrichment of many biological processes related to mitotic cell cycle and chromosome segregation (Fig. 6 E, left panel) and was visualized through enrichment map networks (Fig. 6 E, right panel). Taken together, we have compiled a comprehensive list of phosphorylation sites and potential substrates that are believed to play a significant role in mitotic progression, with TBK1 activity being a key factor (Data S3).

Our analysis could confirm the Aurora kinase family as one of the previously reported off-targets of MRT67307 (Clark et al., 2012). However, our genetic deletion of TBK1-based analysis also revealed a significant inhibition of the Aurora kinase family by TBK1. To validate this, we set out to experimentally validate if TBK1 regulated Aurora A and B kinases to determine the relationship between TBK1 and Aurora A and B kinases. The trans-autophosphorylation sites (T288 AurA, T232 AurB) indicative of activity for Aurora A and B (Hirota et al., 2005) were significantly reduced in our TBK1 KO cells (Fig. 6, F and G). FKBP$^{F36V}$-NAP1 cells treated with dTAG$^v$-1 for 2 h prior to and during G2 release showed a significant reduction in Aurora B phosphorylation but not Aurora A (Fig. 6, H and I; see Discussion). To our knowledge, this is the first time TBK1 has been implicated in regulating Aurora and PAK kinase activity. Besides its direct impact on substrates during mitosis, TBK1's interference with Aurora kinases could lead to dysfunction not only in mitosis but also during cytokinesis. This could be the reason behind the various defects in cell division that we observed.

Based on our GO analyses, it was found that microtubule polymerization and function were significantly represented. When we conducted immunofluorescence staining for p-TBK1, we observed that the signal colocalized with centrosome markers centrin and γ-tubulin foci during mitosis across two independent cell lines (Fig. S5, E and F), reconfirming that TBK1 is activated on centrosomes. However, the p-TBK1 signal also appeared to be localized beyond the centrosomal foci colocalizing with the spindle pole microtubules during metaphase (Fig. S5, E and F). The vicinity of TBK1 activation beyond the centrosomes was also implicated where the area stained by p-TBK1 was significantly reduced due to the loss of NAP1 in HeLa cells (Fig. 4, L and N) and depletion of NAP1 in FKBP$^{F36V}$-NAP1 DLD-1 (Fig. 4, Q and S), suggesting that NAP1 is necessary for TBK1 activation on the spindle poles. These results indicate that TBK1 is involved in both centrosomal organization and function and microtubule processes as noted by the substrates discovered in our phosphoproteomic screen.

### TBK1 phosphorylation of NAP1 at S318 impacts its stability during mitosis

Our phosphoproteomics data indicated that NAP1 is phosphorylated at serine 318 during mitosis by TBK (Fig. 6 B and Fig. S4 D). Although the NAP1 sequence motif at this serine residue did not perfectly match the preferred in vitro TBK1 sequence motif (Fig. 6 D and Fig. S3 B), our data suggested it was a bona fide substrate. First, TBK1 and NAP1 are bound, colocalizing at the centrosomes during mitosis (Fig. 4). Second, TBK1 phosphorylates its own adaptors to modulate their function in other contexts (Heo et al., 2015; Richter et al., 2016; Wild et al., 2011). Third, when comparing the NAP1 sequence motif around S318 against all other 303 kinases, NAP1 still had a high 82.43% kinase preference probability (Data S4). Therefore, we decided to further investigate the connection between NAP1 and TBK1.

First, we tested whether TBK1 affected NAP1 levels. We collected DLD-1 cells at different stages of the cell cycle and checked for NAP1 levels with or without MRT67307 treatment. As expected, the control cells had a significant reduction of NAP1 during mitosis compared with asynchronous conditions; however, MRT67307-treated cells had significantly increased levels of NAP1 (Fig. 7, A and B). Next, we sought to validate NAP1 phosphorylation during mitosis by Phos-tag gel analysis. A higher molecular weight band indicating the presence of a phosphorylated form of NAP1 appeared, which was missing from either TBK1 inhibitor–treated or phosphatase-treated mitotic cell lysates (Fig. 7 C). NAP1 stability was also significantly increased during mitosis in a TBK1 kinase dead mutant (K38A) rescue line (Fig. 7, D and E). The mitotic lysates from TBK1 K38A cells displayed a reduced higher molecular weight NAP1 banding pattern, indicating TBK1 K38A was not able to phosphorylate NAP1 (Fig. 7 F). We then generated an S318A phosphodeficient NAP1 rescue line to test whether TBK1 phosphorylation affected NAP1 stability during mitosis. NAP1 S318A resulted in a significantly higher amount of NAP1 during mitosis as compared with asynchronous conditions (Fig. 7, G and H). Together, these data suggest that activated TBK1 phosphorylates NAP1 on serine 318, which regulates NAP1 degradation (Fig. 7 I).

## Discussion

Our work demonstrates that NAP1/AZI2 is a mitotic and cytokinetic regulatory protein required for normal cell cycle progression. Mechanistically, NAP1 binds to TBK1 at centrosomes to activate TBK1, which subsequently phosphorylates several mitotic and cytokinetic proteins that we identified. NAP1 protein levels during mitosis are tightly regulated by the UPS, where TBK1 phosphorylation of NAP1 at S318 influences its stability (Fig. 7 I).

Previous studies including our own have not extensively characterized all the consequences of cell division upon loss or pharmacological inhibition of TBK1, although these studies reported an increase in the number of multinucleated cells (Maan et al., 2021; Pillai et al., 2015; Sarraf et al., 2019). Previous experiments treating different cancer cell lines with TBK1 inhibitors or siRNAs showed an increase in the presence of supernumerary centrosomes and spindle poles (Maan et al., 2021; Pillai et al., 2015). However, our results find that these defects are more extensive than previously thought, affecting both mitosis and cytokinesis. Mitotic and cytokinetic analyses with cells lacking either TBK1 or NAP1 displayed many of the same phenotypes, which we would expect if NAP1 and TBK1 acted within the same pathway.

The plethora of mitotic and cytokinetic defects found in both NAP1 and TBK1 KO cells led us to perform phospho-proteomic analysis to unbiasedly discover additional TBK1 substrates to explain these diverse types of defects. Comparing the phosphorylation sites to the kinome substrate specificity motifs defined by synthetic peptide libraries, the substrates identified indicated that TBK1 either directly or indirectly affects Aurora A/B and PAK kinase activity as their motifs were highly represented. This was in addition to substrates that are direct TBK1 substrates. HeLa cells lacking TBK1 had a significant reduction in p-Aurora A T288 and p-Aurora B T232. But the FKBP$^{F36V}$-NAP1 knock-in line only displayed reductions in p-Aurora B T232. However, NAP1 loss or depletion also results in cell death, so the differences in cell lines and temporal timing of the loss of the protein need to be considered upon future examination of TBK1's impact on other mitotic kinases. The generation of an inducible conditional TBK1 knockout near-diploid cell line may better distinguish phosphosites in substrates that are compensated for in other cell lines. This also may explain why Plk1 (Kim et al., 2013), which was identified as a substrate in an asynchronous TBK1 KD screen in A549 cells, did not appear as a major hit nor did Plk1 activity appear disrupted in our studies.

Regardless, to our knowledge, this is the first time TBK1 has been implicated in Aurora kinase activity and may explain how the loss of TBK1 and NAP1 is so detrimental. Previously reported substrates, NuMa, CEP170, Plk1, Cdc20, and Cdh1, all function during different stages of mitosis (Kim et al., 2013; Maan et al., 2021; Pillai et al., 2015), but these substrates do not explain all of the cytokinetic defects observed or the increase in binucleated cells in NAP1 and TBK1 KO cells, which can be indicative of cytokinetic failure. Aurora B is important in the formation of the contractile ring and cleavage furrow during cytokinesis (Carmena et al., 2009; Ozlü et al., 2010; Steigemann et al., 2009), so its disruption may explain these defects upon the loss of TBK1

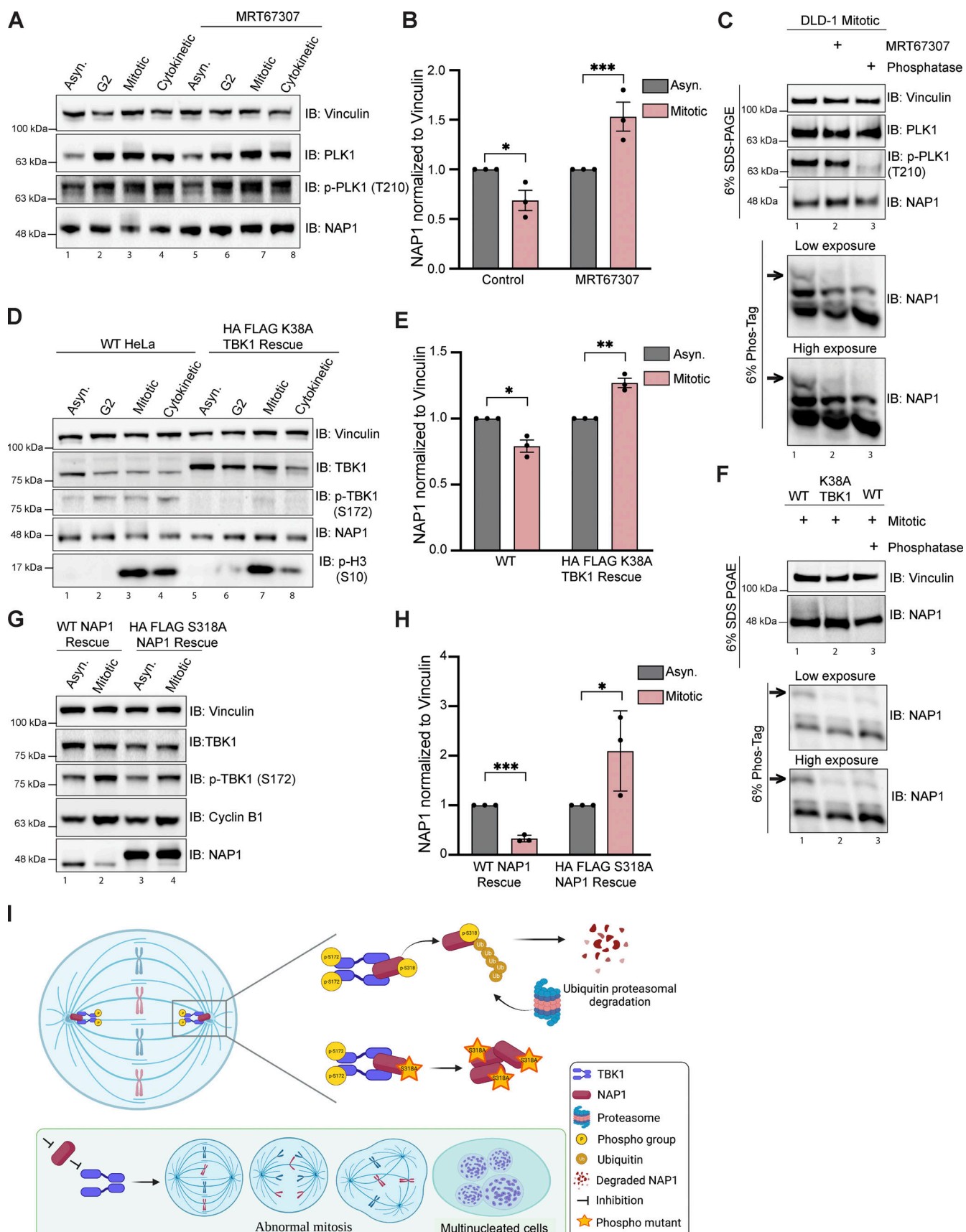

Figure 7. **TBK1 phosphorylation of NAP1 at S318 impacts its stability during mitosis. (A and B)** Representative immunoblots of asynchronous, G2, mitotic, and cytokinetic DLD-1 cells with or without TBK1 inhibitor treatment (MRT67307) for 1 h prior to G2 and throughout release timepoints. Asynchronous

cells were treated for 2 h. Cells were synchronized at G2 using RO-3306. G2 cells were released for ~30–45 min to collect mitotic samples and for ~80–90 min to collect cytokinetic cells. **(B)** Semiquantitative analysis of A normalizing NAP1 protein levels to vinculin in asynchronous and mitotic conditions with or without MRT67307. Error bars indicate ± SEM; n = 3 independent experiments. **(C)** Phos-Tag gel analysis of NAP1 mitotic protein from DLD-1 cells either with MRT67307 for 1 h prior to G2 and throughout release or with 1 h phosphatase treatment. 6% SDS-PAGE gel was run in tandem to ensure equal protein loading. **(D)** Representative immunoblots of asynchronous, G2, mitotic, and cytokinetic cells from WT HeLa and stable TBK1 K38A rescue lines. Cells were synchronized at G2 using RO-3306. G2 cells were released for ~30–45 min to collect mitotic samples and for ~80–90 min to collect cytokinetic cells. **(E)** Semiquantitative analysis of (D) normalizing NAP1 protein levels to vinculin in asynchronous and mitotic conditions in WT HeLa and stable TBK1 K38A rescue lines. Error bars indicate ± SEM; n = 3 independent experiments. **(F)** Phos-Tag gel analysis of NAP1 mitotic protein from HeLa, K38A TBK1 rescue, and HeLa cells treated 1 h with phosphatase. 6% SDS-PAGE gel was run in tandem to ensure equal protein loading. **(G)** Representatives immunoblot analysis of NAP1 in asynchronous and mitotic cells from stable NAP1 and NAP1 S318A rescue lines. Cells were synchronized using RO-3306. **(H)** Semiquantitative analysis of (G) normalizing NAP1 protein levels in asynchronous and mitotic conditions in stable NAP1 rescue and stable NAP1 S318A rescue lines. Error bars indicate ± SEM; n = 3 independent experiments. **(I)** NAP1 activates TBK1 on centrosomes by binding to its adaptor binding C' domain. Activated TBK1 phosphorylates NAP1 on S318 which acts as a signal for ubiquitin proteasomal degradation of NAP1. Loss of either of these centrosomal proteins, NAP1 or TBK1, impairs mitosis and cytokinesis leading to retention of multinucleated cells. One dot = one independent experiment. One-way ANOVA was performed for all statistical analysis. *P < 0.05, **P < 0.01, ***P < 0.001.

and NAP1. Future work will be required to determine the mechanism by which TBK1 affects these other kinases and the careful dissection between bona fide TBK1 substrates and those of other kinases regulated by TBK1.

The *Drosophila* ortholog of TBK1, ik2, is associated with the minus ends of microtubules (Bitan et al., 2010). *ik2* mutant flies display bristle and oocyte abnormalities due to defects in microtubule and cytoskeletal organization due to the inability of mutant ik2 to phosphorylate spn-F and correctly localize ik2 to microtubules (Abdu et al., 2006; Bitan et al., 2010; Dubin-Bar et al., 2008; Lin et al., 2015; Shapiro and Anderson, 2006). Microtubule disruption activates ik2, but dominant negative ik2 reduces microtubule polymerization, suggesting its role in increasing microtubule stability (Barros and Bossing, 2021). We observed p-TBK1 enriched at the spindle poles, as evidenced by its colocalization with γ-tubulin and centrin. However, p-TBK1 was also visible along the spindle microtubules in the region between the spindle poles and the chromosomes. TBK1 phosphorylating NuMa by way of its interaction with CEP170 may be responsible for its association with the spindle poles and correct microtubule tethering (Pillai et al., 2015). However, another paper in the context of Zika infection suggested that CEP63 is involved in TBK1 recruitment to centrosomes identified by γ-tubulin and centrin (Kodani et al., 2022). Considering that there is no mammalian ortholog to *spn-F* and since our analysis revealed several microtubule-associated proteins largely represented in our proteomics data, more work is required to determine specifically where TBK1 is active or inactive on microtubules in different cellular contexts and what downstream consequences this may have to microtubule stability.

We attribute the ability for NAP1 to regulate TBK1 due to its ability to bind to the C' terminal of TBK1. These results are not surprising as it has previously been shown that multiple adaptors bind to this region of TBK1 to facilitate dimerization and trans-autophosphorylation of the S172 site to activate kinase activity (Larabi et al., 2013; Ma et al., 2012). Another possibility is that there is another adaptor not yet identified that can also activate TBK1 during mitosis and participate in regulating its localization or an adaptor that normally does not participate in mitosis can compensate to activate TBK1 in response to the loss of NAP1. TBK1 KO mice are embryonic lethal at day 14.5 (Bonnard et al., 2000). However, the NAP1 knockout mouse is

viable but does display proliferative defects in GM–dendritic cells (Fukasaka et al., 2013). We observed other TBK1 adaptors being phosphorylated during mitosis, but their role is unknown. How TBK1 localizes to centrosomes in the first place is quite intriguing and requires future study.

TBK1 expression is tightly regulated, and our previous data showed abnormal activation and subcellular localization upon overexpression of TBK1 (Sarraf et al., 2019). Elevated expression of TBK1 has been reported in many different types of cancer with high expression correlated with negative patient outcomes (Chen et al., 2017; Uhlen et al., 2017; Wei et al., 2014). This may explain our data showing the tight regulation of NAP1 expression, which displays UPS-dependent degradation from prophase to anaphase, but expression begins to increase and normalize to baseline in the late phases of mitosis and cytokinesis. The regulation of NAP1 protein levels during mitosis may ensure that TBK1 is neither abnormally activated or overactivated nor persists longer than necessary. To understand how the degradation of NAP1 affects TBK1 activation and function during mitosis, manipulation of the E3 ligase responsible for the ubiquitination and degradation of NAP1 is necessary to test this possibility. Future studies for E3 ligase screens are warranted to identify this protein.

We characterized the role of NAP1 and TBK1 during mitosis; other reports indicate that genetic influence and external environmental signaling/stimuli also influence cell division in a TBK1-dependent manner. TBK1's role in cell proliferation was originally described in KRAS mutant cell lines (Barbie et al., 2009), but KRAS mutations interacting with TBK1 are not the sole driver of the proliferation defects (Pillai et al., 2015). Proliferation stimulated by growth factor signaling is reliant on SINTBAD for the activation of TBK1 rather than NAP1 (Zhu et al., 2019), and we did see SINTBAD influence basal p-TBK1 levels in asynchronous conditions. The subcellular localization of TBK1 and its adaptors in these genetic and environmental contexts as well as TBK1 substrates in these situations have yet to be described, making it uncertain if described differences in proliferation are due to changes in mitosis and/or cytokinesis. Both growth factor signaling and KRAS mutations converge on ERK kinase signaling pathways (Katz et al., 2007; Pylayeva-Gupta et al., 2011), and sustained ERK activation throughout G1 is required for entry into the S phase (Jones and Kazlauskas, 2001;

Meloche, 1995; Yamamoto et al., 2006). Thus, it is possible that TBK1 can affect other stages of the cell cycle.

While NAP1 was first identified as an innate immune adaptor (Fujita et al., 2003), our data support a NAP1–TBK1 activation axis occurring at centrosomes ascribing it a new role as a regulator of mitosis and cytokinesis. We have provided mechanistic insight into its function during cell division by providing evidence of how NAP1 activates TBK1 and provided data on how its own stability may be negatively regulated in a TBK1-dependent manner. Most research studies on TBK1-dependent processes such as mitophagy and innate immunity concentrate on asynchronous cell populations or post-mitotic cells ignoring the interplay between cell division and these other processes. However, the ubiquitous nature of TBK1 signaling across different cell types indicates these considerations are warranted. Future work to understand how innate immunity, selective autophagy, and cell division intersect is highly attractive as all three are implicated in human diseases such as cancer.

## Materials and methods

### Cell culture
HeLa and HEK293T cells were maintained in DMEM high glucose medium supplemented with 10% FBS, 2 mM L-glutamine, 10 mM HEPES, 0.1 mM non-essential amino acids, and 1 mM sodium pyruvate. RPE-1 and DLD-1 cells were a kind gift from Dr. Daniela Cimini's lab (Virginia Tech, Blacksburg, VA, USA). RPE-1 cells were maintained in DMEM/F-12 medium supplemented with 10% FBS. DLD-1 cells were maintained in RPMI-1640 medium with 10% FBS. Penta KO and DKO HeLa cells were a kind gift from Dr. Richard J. Youle's lab. THP1-Lucia ISG cells (Invivogen) were maintained in RPMI-1640 medium with 10% FBS, 2 mM L-glutamine, 25 mM HEPES, 100 µg/ml Normocin, and Pen-Strep (100 U/ml–100 µg/ml). Cells were routinely tested for mycoplasma contamination by PCR (Southern Biotech).

### Antibodies
The following antibodies were used for this study: rabbit anti-TBK1/NAK (#3504S/#14590S; Cell Signaling Technology [CST]), rabbit anti-pTBK1 (Ser172; #5483S; CST), rabbit anti-pTBK1 Alexa Fluor 488 or 647 conjugated (Ser172; #14586/#14590; CST), rabbit anti-NAP1/AZI2 (#15042-1-AP; Proteintech), rabbit anti-GAPDH (G9545; Sigma-Aldrich), mouse anti-Vinculin (#700062; Invitrogen), rabbit anti-p-Histone H3 (S10; #53348S; CST), rabbit anti-SINTBAD (#8605S; CST), rabbit anti-TANK (#2141S; CST), mouse anti-FLAG M2 (#F1804; Sigma-Aldrich), mouse anti-p62 (#H00008878-M01; Abnova), mouse anti-NBR1 (#H00004077-M01; Abnova), rabbit anti-TAX1BP1 (HPA024432; Sigma-Aldrich), rabbit anti-optineurin (#10837-1-AP; Proteintech), rabbit anti-NDP52 (#60732; CST), mouse anti-α-Tubulin (#T6074; Sigma-Aldrich) rabbit anti-α-Tubulin (#2144S; CST; ab52866; Abcam), mouse anti-GFP (Cat#11814460001; Roche), rabbit anti-IKKε (#2905S; CST), rabbit anti-pIKKε (S162; #8766S, CST), mouse anti-HA.11 (#901513; BioLegend), rabbit anti-IRF3 (#4302; CST), rabbit anti-p-IRF3 (S386; ab76493; Abcam), rabbit anti-Plk1 (#4513T; CST), rabbit anti-p-Plk1 (Thr210; # 5472T; CST), human anti-CREST (#15-234; Antibodies Inc.), rabbit anti-CDK1/

CDC2 (#77055; CST), rabbit anti-pCDK1/CDC2 (Tyr15; #4539; CST), rabbit anti-LC3 (NB600-1384; Novus; #2775S; CST), mouse anti-γ-tubulin (#5326; Sigma-Aldrich), rabbit anti-Aurora A (#14475; CST), rabbit anti-Aurora B (#3094T; CST), rabbit anti-Aurora A/B (T288/T232; #2914T; CST), mouse anti-Centrin (#04-1624; EMD Millipore), and mouse anti-cyclin B1 (#SC-245; Santa Cruz).

### Additional chemicals
The following chemicals were used for this study: 10 µg/µl cycloheximide (AC357420010; Thermo Fisher Scientific), 10 µm MG132 (NC9937881; Thermo Fisher Scientific), 1 µM TAK243 (30108; Cayman Chemical), 50 µM chloroquine diphosphate (C2301100G; Thermo Fisher Scientific), 3 µM MRT67307 (inh-mrt; Invivogen), 5 mg/ml LPS (eBioscience), 1 mg/ml poly I:C (Thermo Fisher Scientific), 5U FastAP Thermosensitive Alkaline Phosphatase (EF0651; Thermo Fisher Scientific), 1 µM dTAG$^V$-1 (Toris), and 10 µM BI605906 (50-203-0195; Thermo Fisher Scientific).

### Plasmids and constructs
To generate NAP1 rescue lines, NAP1 cDNA was cloned into pDONR223 and transferred into the pHAGE-N′-FLAG-HA-IRES-puro, pHAGE-N′-EGFP-Gaw-IRES-Blast, or pHAGE-C′-EGFP-Gaw-IRES-Blast vectors using LR recombinase (Invitrogen). The pHAGE-N′-FLAG-HA-TBK1 cloning has been described previously (Sarraf et al., 2019) and deposited to #131791; Addgene. The Lenti6–H2B–mCherry construct was a gift from Torsten Wittmann (UCSF, San Francisco, CA, USA; plasmid #89766; Addgene). The pHAGE-N′-FLAG-HA-TBK1 K38A construct was a kind gift from Dr. Richard Youle's lab (National Institutes of Health, Bethesda, MD, USA). The following site-directed mutagenesis primers were used to generate mutant constructions: NAP1 Δ230–270: 5′-TTCATCAAGTGCAGTTTTGTATATGGATCC GTTTGTTTGGCTTTC-3′, 5′-GAAAGCCAAACAAACGGATCCATA TACAAAACTGCACTTGATGAA-3′; TBK1 Δ C′ terminus: 5′-GGG GACAAGTTTGTACAAAAAAGCAGGCTTCGAGGAGATAGAACC ATGATGCAGAGCACTTCTAATCATCTG-3′, 5′-GGGGACCACTTT GTACAAGAAAGCTGGGTCCTACTATATCCATTCTTCTGACTT ATT-3′. NAP1 S318A: 5′-AATCCTCCAAGCATGGACAGACA-3′, 5′-GCTTTCTCTGATAAAACCTTTACATC-3′. All constructs were confirmed by DNA sequencing and deposited on Addgene.

### Retrovirus and lentivirus generation
Dishes were coated with 50 µg/ml poly-d-lysine (Sigma-Aldrich), and HEK293T cells were plated at 70–80% confluency before transfection. Lentiviral helpers and constructs were transfected using XtremeGENE 9 (Roche) according to the manufacturer's instructions at a 1:3 ratio. 24 h after transfection, the media was changed. Infectious media containing the virus was collected 40 h later and filtered with a 0.45-µm PES membrane filter (Millipore). Viral purification was performed using an Optima MAX-XP ultracentrifuge (Beckman) and spinning media at 100,000 × g for 2 h at 4°C. Viral pellet was resuspended in sterile PBS, and the tier was quantified using qPCR Lentivirus Titer Kit (abm) according to the manufacturer's directions. Live-filtered virus was used to transduce cells with polybrene (10 µg/ml; Sigma-Aldrich).

## CRISPR knockout cell line generation

In brief, CRISPR design was aided by publicly available software provided by MIT. CRISPR oligos for NAP1 were 5′-AAACCAGCT GGAGGAGTTCTACTTC-3′ and 5′-CACCGAAGTAGAACTCCT CCAGCTG-3′. Primers were annealed with Phusion DNA polymerase (Thermo Fisher Scientific) using the following conditions: 98°C for 1′, two to three cycles of 98°C for 10″, 53°C for 20″, 72°C for 30″, and 72°C for 5′. The annealed primers were cloned into the linearized gRNA vector, which was a gift from Dr. Feng Zhang (MIT, Cambridge, MA, USA; plasmid #62988; Addgene) using the Gibson Assembly Cloning Kit (NEB). HeLa cells were cotransfected using XtremeGENE 9 (Roche) using the above CRISPR plasmid. Cells were selected by puromycin (1 mg/ml) and serially diluted into 96-well plates to select for single colony clones. DNA was extracted from individual clones using the Zymo gDNA Isolation Kit and genotyped/sequenced using the following primers: exon 4 F 5′-GAAGCGAATGACATCTGCA-3′, exon 4 R 5′-CCTCTTCTGCTTCATCACAACCT-3′.

## shRNA cell line generation

pLKO.1 puro was a gift from Dr. Bob Weinberg (MIT; plasmid #8453; Addgene) digested with AgeI and EcoRI for 4 h at 37°C. The digested plasmid was excised, and the gel was purified with GeneJET gel extraction kit (Thermo Fisher Scientific). Oligos were designed for the following target sequences and annealed (NAP1: 5′-CCGGCCACTGCATTACTTGGATCAACTCGAGTTGAT CCAAGTAATGCAGTGGTTTTTG-3′; 5′-AATTCAAAAACCACT GCATTACTTGGATCAACTCGAGTTGATCCAAGTAATGCAGTGG-3′, TANK: 5′-CCGGCCTCAAAGTCTACGAGATCAACTCGAGTT GATCTCGTAGACTTTGAGGTTTTTG-3′; 5′-AATTCAAAAACC TCAAAGTCTACGAGATCAACTCGAGTTGATCTCGTAGACTTT GAGG-3′, SINTBAD: 5′-CCGGCCTCTGCCTTTCTGTTCTTAACT CGAGTTAAGAACAGAAAGGCAGAGGTTTTTG-3′; 5′-AATTCA AAAACCTCTGCCTTTCTGTTCTTAACTCGAGTTAAGAACAGA AAGGCAGAGG-3′). Annealed oligos were ligated into the digested vector with T4 ligase (NEB), and colonies were screened by sequencing. Scramble shRNA was a gift from Dr. David Sabatini (MIT; plasmid #1864; Addgene). Cells were selected using 1 mg/ml puromycin.

## FKBP12$^{F36V}$ degradation tag-NAP1 cell line generation

The cell line was generated as previously described (Damhofer et al., 2021). For the donor vector, the BSD-P2A-2xHA-FKBP12$^{F36V}$ and backbone pCRIS-PITCH cassettes were generated using PCR. The NAP1 homology sequences for N terminal tagging were synthesized as a gBlock gene fragment (IDT) and they were assembled using NEBuilder HiFi DNA Assembly Master Mix (E2621; NEB) according to the manufacturer's instructions. For guide RNA expression targeting NAP1, pX459 was cloned as previously described. Briefly, guides targeting the N-terminal of NAP1 were designed using the CHOPCHOP website (https://chopchop.cbu.uib.no/). Oligonucleotides (5′-CACCGAACAGTT GTCATGGATGCAC-3′, 5′-AAACGTGCATCCATGACAACTGTT-3′) from IDT were annealed and inserted into the pX459 plasmid after BbsI (NEB) digestion. To develop the FKBP12$^{F36V}$ degradation tag-NAP1 DLD-1 cell, DLD-1 cells were cotransfected using Lipofectamin 3000 (Invitrogen) using the above donor vector

and pX459 plasmid targeting NAP1. Cells were selected by puromycin (1 mg/ml) and diluted into 96-well plates to select for single colony clones. gDNA was extracted from individual clones using lysis buffer (25 mM KCl, 5 mM Tris HCl pH 8.0, 1.25 mM MgCl2, 0.2% NP40, 0.2% Tween-20, 0.4 µg/ml Proteinase K) and heated at 65°C for 10 min and 98°C for 5 min incubation. PCR for genotyping was conducted using the following primers: 5′-AAG AACTTTTGAAAATTTATAAATTGAG-3′, 5′-GAAAAATATTTGGA ATATAACTCCAAG-3′.

## Cell synchronization

Nocodazole treatment: Cells were incubated with 1 µg/ml nocodazole (Sigma-Aldrich) containing medium to synchronize at prometaphase for 16 h and collected for further experiments.

R0-3306 treatment: To synchronize at G2, cells were reversibly incubated with 9 µM RO-3306 (TCI America) containing medium for 20 h and collected at the following time points corresponding to their respective cell cycle stages when released in normal growth medium. 0 h—G2; 1 h—M (metaphase); 7 h—G1. The mitotic shake was employed to obtain the maximum number of mitotic cells. The remaining cells on the cell culture plate were discarded to eliminate interphase cells from mitotic sample collection.

## Immunocytochemistry

Cells for immunofluorescence imaging were plated in six-well cell culture plates (Corning Incorporated) on glass coverslips. Cells were fixed with 4% PFA for 10 min and permeabilized with 0.1% Triton X-100 for 10 min followed by blocking with 10% BSA and 5% NGS for 45 min at RT. Cells were incubated with primary antibodies (diluted in 5% BSA and 2.5% NGS) overnight at 4°C followed by washing with 1X PBS and incubated with AlexaFluor (Thermo Fisher Scientific) conjugated secondary antibodies in the dark for 1 h. Following the washing step, the cells were stained with 1 µg/ml DAPI for 5 min (Thermo Fisher Scientific) and mounted on the slides using Fluoromount (Southern-Biotech). Imaging was carried out using a Nikon C2 confocal microscope.

For p-TBK1 centrosomal intensity and area analysis, a different permeabilization method was performed. Cells were permeabilized with 100% ice-cold methanol in the refrigerator for 10 min.

## Cell viability assay for growth curve

Approximately, 200–400 cells were plated (four-wells/genotype) in white-coated 96-well plates (Brand Tech Scientific) in growth media. The cell growth curve was obtained by CellTiter-Glo Luminescent Cell Viability Assay (Promega) using a luminescence reader every 24 h. The mean cell number corresponding to the luminescence on each day was normalized to the first day in the graph.

## Cell defects analysis

Fixed cells were immunostained with α-tubulin for cytoskeleton marker, CREST for kinetochore marker, and DAPI for the nuclear counter stain. For the mitotic index, nuclear defects cell count (binucleation and multinucleation) and cytokinetic cell count random fields of view were captured for each genotype to sample ~1,000 cells for each biological replicate (n = 3). Mitotic

cells were identified by chromosome condensation and kinetochore staining and verified by α-tubulin morphology. For mitotic defects analysis, random fields of view were captured for each genotype to sample ~50 mitotic cells per biological replicate ($n$ = 3; around 150 mitotic cells per genotype), and for cytokinetic defects analysis, random fields of view were captured for each genotype to sample ~30 cytokinetic cells per biological replicate ($n$ = 3; around 90 cytokinetic cells per genotype).

All mitotic defects analyses were performed by following the guidelines (Baudoin and Cimini, 2018). We observed the following spindle and segregation abnormalities from the z-projection of the confocal images. Monopolar spindle: prometaphase cells with just one spindle pole and a single microtubule aster. Multipolar spindle: prometaphase and metaphase cells with more than two spindle poles. Splayed/unfocused spindle: prometaphase and metaphase spindle poles with multiple focal points instead of a distinct one. Multipolar anaphase/telophase: the cells with multipolar spindles that are already in the chromosome segregation phase. Although this defect could be grouped with multipolar spindle, creating a different mitotic defect category distinguishes the defects observed in the prometaphase/metaphase from the anaphase/telophase. Unequal nuclear division anaphase/telophase: anaphase and telophase cells with an unequal mass of segregating chromosomes. Acentric fragments/chromatin bridge in anaphase/telophase: cells in anaphase or telophase where either a DNA stretch is connected between the segregating chromosomes or a fragment of chromosome is positioned away from the main mass of segregating chromosomes.

## Time-lapse imaging

For time-lapse imaging, dTAG DLD-1 cells were transiently transfected with H2B–mCherry for 48 h with XtremeGENE 9 (Roche). Cells were grown in the chamber slides (#155379PK; Thermo Fisher Scientific). Prior to the live cell imaging setup, the cells were pretreated with either DMSO or 1 µM dTAG for 1 h. All time-lapse experiments were performed on a Nikon C2 microscope (Nikon Instruments, Inc.) equipped with an electrically heated chamber for the Nikon motorized stage, stage top incubator's controller for application with premixed gases. It includes a thermal controller, the humidity module, and a single-channel gas flow rate regulator. OKO3 gas mixer with an external air pump. Also, it is outfitted with brightfield, Kinetix CMOS camera, and fluorescent channels DAPI, GFP, and Texas Red. At the time of imaging, at least 10 fields of view were selected for each biological replicate to capture the maximum number of cells. The cells were followed upto 4–5 h with either DMSO for the untreated group or 1 µM dTAG for the treated group. Images were acquired every 3 min through a 40× brightfield and fluorescent objective for the duration of the experiment.

For the analysis of time-lapse videos on the Fiji ImageJ software, only cells expressing H2B–mCherry, which went into mitosis after the experimental setup, were selected. The duration of early mitotic stages was quantified only in the visibly normal near diploid cells. All binucleated and multinucleated cells were excluded from the analysis. The duration of early mitotic stages was quantified using the following parameters. Prophase—from the beginning of the visible chromosome

structure with slight condensation within the nuclear perimeter until nuclear envelope breakdown; prometaphase—nuclear envelope breakdown until metaphase plate formation; metaphase: from the metaphase plate formation until the beginning of chromosome segregation. All cells were followed through cytokinesis for the identification of any mitotic or cytokinetic defects. A total of 100 cells from three biological replicates per treatment were observed for all analyses.

## Cell collection and treatment for phospho-proteomics

WT HeLa and two independent TBK1 CRISPR knockout lines (TBK1 KO clone 2 and clone 4) were treated with 9 µM RO-3306 to synchronize at G2. Cells were released in normal growth medium for an hour to collect ~20 M mitotic cells per sample (using mitotic shake). To account for the off-target gene editing effect, additional groups were added with TBK1 inhibitor treatment. For this, WT HeLa and both TBK1 KO clones were also treated 9 µm with R0-3306 for 19.5 h 30 min prior to the G2 wash, the cells were treated with 3 µm MRT67307 along with 9 µm R03306. After G2 wash the cells were released in normal growth medium with 3 µm MRT67307 for 1 h to collect ~20 M mitotic cells per sample. Four biological replicates were collected for each group (Fig. 1 A).

## Proteomics—cell lysis and protein digestion

At the indicated times, cells were washed twice with ice-cold PBS and snap-frozen. Cell pellets were lysed in lysis buffer (25 mM EPPS pH 8.5, 8 M Urea, 150 mM NaCl, phosphatase, and protease inhibitor cocktail [in-house]) to produce whole-cell extracts. Whole-cell extracts were sonicated and clarified by centrifugation (16 000 × $g$ for 10 min at 4°C) and protein concentrations were determined by the Bradford assay. Protein extracts (3 mg) were subjected to disulfide bond reduction with 5 mM TCEP (room temperature, 10 min) and alkylation with 25 mM chloroacetamide (room temperature, 20 min). Methanol–chloroform precipitation was performed prior to protease digestion. In brief, four parts of neat methanol were added to each sample and vortexed, one part chloroform was then added to the sample and vortexed, and finally three parts water was added to the sample and vortexed. The sample was centrifuged at 6,000 rpm for 5 min at room temperature and subsequently washed twice with 100% methanol. Samples were resuspended in 100 mM EPPS pH 8.5 containing 6 M Urea and digested at 37°C for 2 h with Lys-C at a 100:1 protein-to-protease ratio. Samples were then diluted to 0.5 M Urea with 100 mM EPPS pH 8.5 solution, trypsin was then added at a 100:1 protein-to-protease ratio and the reaction was incubated for 6 h at 37°C. Digestion efficiency of a small aliquot was tested, and samples were acidified with 0.1% trifluoroacetic acid (TFA) final and subjected to C18 solid-phase extraction (SPE; Sep-Pak, Waters).

## Proteomics—Fe2+-NTA phosphopeptide enrichment

Phosphopeptides were enriched using Pierce High-Select Fe2+-NTA phosphopeptide enrichment kit (A32992; Thermo Fisher Scientific) following the provided protocol. In brief, dried peptides were enriched for phosphopeptides and eluted into a tube containing 25 µl 10% formic acid (FA) to neutralize the pH of the elution buffer and dried down. The unbound peptides (flow-

through) and washes were combined and saved for total proteome analysis.

## Proteomics—tandem mass tag labeling

Proline-based reporter isobaric Tandem Mass Tag (TMTpro) labeling of dried peptide samples resuspended in 100 mM EPPS pH 8.5 was carried out as follows. For total proteome analysis (50 mg of flow through peptide) and for phosphopeptide proteomics, (desalted, eluted peptides from phospho-enrichment step), 10 µl of a 12.5 µg/µl stock of TMTpro reagent was added to samples, along with acetonitrile to achieve a final acetonitrile concentration of ∼30% (vol/vol). Following incubation at room temperature for 1 h, the labeling efficiency of a small aliquot was tested for each set (total proteome and phospho-proteome), and the reaction was then quenched with hydroxylamine to a final concentration of 0.5% (vol/vol) for 15 min. The TMTpro-labeled samples were pooled together at a 1:1 ratio. The total proteome sample and phospho-proteome sample were vacuum centrifuged to near dryness and subjected to C18 solid-phase extraction (SPE; 50 mg, Sep-Pak, Waters).

## Proteomics—off-line basic pH reversed-phase (BPRP) fractionation

A dried TMTpro-labeled sample was resuspended in 100 µl of 10 mM $NH_4HCO_3$ pH 8.0 and fractionated using basic pH reverse phase HPLC (Wang et al., 2011). Briefly, samples were offline fractionated over a 90-min run into 96 fractions by high pH reverse-phase HPLC (LC1260; Agilent) through an (1) aeris peptide xb-c18 column (250 × 3.6 mm; Phenomenex) for total proteome, (2) kinetex EVO-c18 column (150 × 2.1 mm; Phenomenex) for phosphor-proteome, with mobile phase A containing 5% acetonitrile and 10 mM $NH_4HCO_3$ in LC-MS grade $H_2O$, and mobile phase B containing 90% acetonitrile and 10 mM $NH_4HCO_3$ in LC-MS grade $H_2O$ (both pH 8.0). The 96 resulting fractions were then pooled in a non-continuous manner into 24 fractions (as outlined in Fig. S5 of Paulo et al. [2016a]) used for subsequent mass spectrometry analysis. Fractions were vacuum-centrifuged to near dryness. Each consolidated fraction was desalted via StageTip, dried again via vacuum centrifugation, and reconstituted in 5% acetonitrile, and 1% formic acid for LC-MS/MS processing.

## Proteomics—total proteomics analysis using TMTpro

Mass spectrometry data were collected using an Orbitrap Eclipse Tribrid mass spectrometer (Thermo Fisher Scientific) coupled to an UltiMate 3000 RSLCnano system liquid chromatography (LC) pump (Thermo Fisher Scientific). Peptides were separated on a 100-µm inner diameter microcapillary column packed in-house with ∼40 cm of HALO Peptide ES-C18 resin (2.7 µm, 160 Å; Advanced Materials Technology) with a gradient consisting of 5–21% (0–85 min), 21–28% (85–110 min; ACN, 0.1% FA) over a total 120 min run at ∼500 nl/min. For analysis, we loaded 1/10 of each fraction onto the column. Each analysis used the Multi-Notch MS³-based TMT method (McAlister et al., 2014) to reduce ion interference compared with MS² quantification (Paulo et al., 2016b), combined with the FAIMS Pro Interface (using previously optimized three CV parameters for TMT multiplexed

samples (Schweppe et al., 2019) and combined with newly implemented Real Time Search analysis software (Erickson et al., 2019; Schweppe et al., 2020a)). The scan sequence began with an MS¹ spectrum (Orbitrap analysis; resolution 120,000 at 200 Th; mass range 400–1,500 m/z; automatic gain control [AGC] target 4×10⁵; maximum injection time 50 ms). Precursors for MS² analysis were selected using a cycle type of 1.25 s/CV method (FAIMS CV = −40/−60/−80). MS² analysis consisted of collision-induced dissociation (quadrupole ion trap analysis; rapid scan rate; AGC 1.0×10⁴; isolation window 0.5 Th; normalized collision energy [NCE] 35; maximum injection time 35 ms). Monoisotopic peak assignment was used, and previously interrogated precursors were excluded using a dynamic window (180 s ±10 ppm). Following the acquisition of each MS² spectrum, a synchronous-precursor-selection (SPS) API-MS³ scan was collected from the top 10 most intense ions b or y ions matched by the online search algorithm in the associated MS² spectrum (Erickson et al., 2019; Schweppe et al., 2020a). MS³ precursors were fragmented by high-energy collision-induced dissociation (HCD) and analyzed using the Orbitrap (NCE 45; AGC 2.5 × 10⁵; maximum injection time 200 ms, resolution was 50,000 at 200 Th). The closeout was set at two peptides per protein per fraction so that MS³s were no longer collected for proteins having two peptide-spectrum matches (PSMs) that passed quality filters (Schweppe et al., 2020a).

## Proteomics—phosphoproteomics analysis using TMTpro

Mass spectrometry data were collected using an Orbitrap Eclipse Tribrid mass spectrometer (Thermo Fisher Scientific) coupled to an UltiMate 3000 RSLCnano system liquid chromatography (LC) pump (Thermo Fisher Scientific). Peptides were separated on a 50-cm µPAC column (PharmaFluidics) with a gradient consisting of 3–18% (0–85 min), 18–25% (85–110 min; ACN, 0.1% FA) over a total 125 min run at ∼250 nl/min. For analysis, we loaded half of each fraction onto the column. Each analysis used the FAIMS Pro Interface (using previously optimized three CV parameters for TMTprolabeled phosphopeptides [Schweppe et al., 2020b]) to reduce ion interference. The scan sequence began with an MS¹ spectrum (Orbitrap analysis; resolution 120,000 at 200 Th; mass range 400–1,500 m/z; automatic gain control [AGC] target 4 × 10⁵; maximum injection time 50 ms). Precursors for MS² analysis were selected using a cycle type of 1.25 s/CV method (FAIMS CV = −40/−60/−80). MS² analysis consisted of high-energy collision-induced dissociation (HCD; Orbitrap analysis; resolution 50,000 at 200 Th; isolation window 0.5 Th; normalized collision energy [NCE] 38; AGC 2 × 10⁵; maximum injection time 86 ms). Monoisotopic peak assignment was used and previously interrogated precursors were excluded using a dynamic window (120 s ± 10 ppm).

## Proteomics—data analysis

Mass spectra were processed using a Comet-based (2020.01 rev. 4) software pipeline (Eng et al., 2013). Spectra were converted to mzXML and monoisotopic peaks were reassigned using Monocle (Rad et al., 2021). MS/MS spectra were matched with peptide sequences using the Comet algorithm (Eng et al., 2013) along with a composite sequence database including the Human

Reference Proteome (2020-01—SwissProt entries only) UniProt database, as well as sequences of common contaminants. This database was concatenated with one composed of all protein sequences in the reversed order. Searches were performed using a 50 ppm precursor ion tolerance for analysis. For total proteomic analysis, the recommended product ion parameters for ion trap ms/ms were used (1.0005 tolerance, 0.4 offset [mono masses], theoretical fragment ions = 1). For phosphoproteomics analysis, the recommended product ion parameters for high-resolution ms/ms were used (0.02 tolerance, 0.0 offset [mono masses], theoretical fragment ions = 1). TMTpro tags on lysine residues and peptide N termini (+304.207 D) and carbamidomethylation of cysteine residues (+57.021 D) were set as static modifications, while oxidation of methionine residues (+15.995 D) was set as a variable modification. For the phosphorylation dataset search, phosphorylation (+79.966 D) on serine or threonine wase set as additional variable modifications. Peptide-spectrum matches (PSMs) were adjusted to a 1% false discovery rate (FDR; Elias and Gygi, 2007). PSM filtering was performed using a linear discriminant analysis (Huttlin et al., 2010) while considering the following parameters: Comet Log Expect, Diff Seq. Delta Log Expect, missed cleavages, peptide length, charge state, and precursor mass accuracy. For protein-level comparisons, PSMs were identified, quantified, and collapsed to a 1% peptide false discovery rate (FDR) and then collapsed further to a final protein-level FDR of 1% (Savitski et al., 2015). Moreover, protein assembly was guided by principles of parsimony to produce the smallest set of proteins necessary to account for all observed peptides. For TMTpro-based reporter ion quantitation, we extracted the summed signal-to-noise (S:N) ratio for each TMTpro channel and found the closest matching centroid to the expected mass of the TMT reporter ion (integration tolerance of 0.003 D). Reporter ion intensities were adjusted to correct for the isotopic impurities of the different TMTpro reagents according to manufacturer specifications. Proteins were quantified by summing reporter ion signal-to-noise measurements across all matching PSMs, yielding a "summed signal-to-noise" measurement. For total proteome, PSMs with poor quality, $MS^3$ spectra with eight or more TMTpro reporter ion channels missing, or isolation specificity <0.7, or with TMT reporter summed signal-to-noise ratio that were <160 or had no $MS^3$ spectra were excluded from quantification. For phosphoproteome, PSMs with poor quality, $MS^3$ spectra with 12 or more TMT reporter ion channels missing, or isolation specificity <0.8, or with TMT reporter summed signal-to-noise ratio that were <160 or had no $MS^3$ spectra were excluded from quantification. Phosphorylation site localization was determined using the AScorePro algorithm (Beausoleil et al., 2006; Gassaway et al., 2022). AScore is a probability-based approach for high-throughput protein phosphorylation site localization. Specifically, a threshold of 13 corresponded to 95% confidence in site localization.

Protein or peptide quantification values were exported for further analysis in Microsoft Excel, GraphPad Prism, R package, and Perseus (Tyanova et al., 2016). Each reporter ion channel was summed across all quantified proteins and normalized assuming equal protein loading of all samples. Phospho peptides were normalized to the protein abundance value (when

available), and then normalization of the dataset was performed using PhosR package (Kim et al., 2021). Gene Ontology enrichment analyses were performed with R package Cluster-Profiler (4.0; Wu et al., 2021).

Supplemental datasets list all quantified proteins as well as the associated TMT reporter ratio to control channels used for quantitative analysis.

### Serine/threonine kinase predictions
Kinase predictions were based on experimental biochemical data of their substrate motifs. We utilized synthetic peptide libraries containing 198 peptide mixtures that explored amino acid preference up to five residues N-terminal and C-terminal to the phosphorylated Ser/Thr to determine the optimal substrate sequence specificity for recombinant Ser/Thr kinases. In total, 303 kinases were profiled. Their motifs were quantified into position-specific scoring matrices (PSSMs) and then applied computationally to score phosphorylation sites based on their surrounding amino acid sequences. These PSSMs were ranked against each site to identify the most favorable kinases. This work is in preparation (Johnson et al., 2022 *Preprint*).

### The kinase library enrichment analysis
The phosphorylation sites detected in this study were scored by all the characterized kinase PSSMs (303 S/T kinases), and their ranks were determined (Johnson et al., 2022 *Preprint*). For every non-duplicate, singly phosphorylated site, kinases that ranked within the top 15 out of the 303 S/T total kinases were considered as biochemically predicted kinases for their respective phosphorylation site. For assessing kinase motif enrichment, we compared the percentage of phosphorylation sites for which each kinase was predicted among the downregulated/upregulated phosphorylation sites (sites with |$\log_2$ fold change| ≥1 and with FDR ≤0.1) with the percentage of biochemically favored phosphorylation sites for that kinase within the set of unregulated sites in this study (sites with |$\log_2$ fold change| <1 and with FDR >0.1). Statistical significance was determined using a one-sided Fisher's exact test, and the corresponding P values were adjusted using the Benjamini–Hochberg procedure. Kinases that were significant (adjusted P value 0.1) for both upregulated and downregulated analysis were excluded from downstream analysis. Then, for every kinase, the most significant enrichment side (upregulated or downregulated) was selected based on the adjusted P value and presented in the scatterplot.

### Western blots
For immunoblotting, cells were lysed using 1X RIPA buffer (Thermo Fisher Scientific Pierce RIPA Buffer) containing 1X protease/phosphatase inhibitor cocktails (Thermo Fisher Scientific Halt). Protein concentration was quantified using the DC Protein Assay Kit (Bio-Rad). Cell lysates were boiled for 15 min with 2X LDS buffer containing 50 mM DTT and 20 μg of protein lysates were resolved by 4–12% Bis-Tris gels and transferred to PVDF membranes. Blots were blocked using 5% non-fat powdered milk in 1X TBST (150 mM NaCl, 20 mM Tris, pH 8.0, 0.1% Tween 20). Primary and secondary antibody incubations were carried out in 2.5% non-fat powdered milk in 1X

TBST overnight at 4°C and 1 h at room temperature, respectively. Blots were exposed using Clarity Western ECL Substrates (Bio-Rad), ECL Select Western Blotting Detection Reagent (GE Healthcare), or SuperSignal West Femto Maximum Sensitivity Substrate (Thermo Fisher Scientific) and detected by the ChemiDoc Imaging System (Bio-Rad). Protein bands from the Western blot were quantified using the volume toolbox of Image Lab image acquisition and analysis software (Bio-Rad). For Western blot quantification graphs, the signals from the protein of interest were normalized with that of the loading control.

### PhosTag gels
For protein phosphorylation analysis, 6% Supersep Phos-tag gels (192-17401; Wako Chemicals) were used. These gels were run in 1% Tris Glycine and SDS buffer for ∼2 h. To increase the transfer efficiency, the phostag gels were soaked for 10 min (three times each) in the general transfer buffer containing 5 mM EDTA. After washing off the EDTA from the gels using a normal transfer buffer, conventional Western blot transfer, blocking, and antibody incubation steps were followed. For lysates that underwent phosphatase treatment, cells were lysed using 1X RIPA buffer (Thermo Fisher Scientific Pierce RIPA Buffer) containing 1X protease inhibitor cocktail-EDTA free (Sigma-Aldrich). Approximately, 200–250 mg of protein lysate was treated with 5 U of FastAP alkaline phosphatase (Thermo Fisher Scientific) for 1 h at 37°C.

### Immunoprecipitation
Cells were lysed with the following lysis buffer: 50 mM Tris/HCl pH 7.5, 150 mM NaCl, 1 mM EGTA, 1 mM EDTA, 0.5 (vol/vol) NP-40, 1 mM sodium orthovanadate, 50 mM NaF, 5 mM sodium pyrophosphate, 0.27 M sucrose, 10 mM Na 2-glycerophosphate, 0.2 mM phenylmethylsulphonyl fluoride, 1× protease/phosphatase inhibitor cocktail (Pierce), and 50 mM iodoacetamide. Cells were incubated for 20 min end over end at 4°C then spun at 16,000 × $g$ for 15 min at 4°C. The supernatant was measured using the DC protein assay (Bio-Rad). 500 µg of protein was incubated on magnetic beads (FLAG M2 beads [Sigma-Aldrich], GFP beads [Chromotek]) end over end at 4°C for 2 h. For endogenous IPs, the supernatant was precleared on TrueBlot Protein G magnetic beads (Rockland) for 30 min at 4°C. 1.2 mg of protein was incubated with 2 µL of TBK1 antibody for 1 h at 4°C (#3013S; Cell Signaling). Magnetic beads were added for 1 h at 4°C. 2X LDS with 50 mM DTT was used to elute protein off the beads, and pH was restored with NaOH.

### Transfection
HEK293T cells were plated on poly-d-lysine coated dishes and reverse transfected with 1:1 jetOPTIMUS transfection reagent (Polyplus). Cells were treated the next day with 100 ng/ml nocodazole (Sigma-Aldrich) and collected 16 h later. HeLa cells were reverse transfected with 1:3 or 1:6 XtremeGENE 9 (Roche) transfection reagent according to the manufacturer's instructions.

### qPCR
RNA was isolated using Trizol Reagent (Ambion Life Technologies) and converted to cDNA using iScript (Bio-Rad) as per the manufacturer's directions. SYBR Green Supermix (Bio-Rad), 20 ng of cDNA, and 0.4 µM of each primer set were mixed in a 10 µl RT-PCR reaction that was run on the CFX96 System (Bio-Rad). The primers that were used span exons: Actin F: 5′-CCCGCCGCC AGCTCACCAT-3′, R: 5′-CGATGGAGGGGAAGACGGCCC-3′. TANK F: 5′-AGCAAGGAGTCTTGGCAGTC-3′, R: 5′-GCACTGTGT TTCAGTTGCAGT-3′. SINTBAD F: 5′-ACCAGTTCCAGCATGAGTT ACA-3′, R: 5′-TCTCCCTCAGCTCTGTCTCC-3′. AZI2 F: 5′-AGG TGGAAACTCAGCAGGTG-3′, R: 5′-ATGGATCCGTTTGTTTGG CT-3′. IL-6 F: 5′-AGCCACTCACCTCCTCAGAACGAA-3′, R: 5′-AGTGCCTCTTTGCTGCTTTCACAC-3′. TNF-α F: 5′-TCAATCGGC CCGACTATCTC-3′, R: 5′-CAGGGCAATGATCCCAAAGT-3′. IL-10 F: 5′-AAGACCCAGACATCAAGGCG-3′, R: 5′-CAGGGAAGAAAT CGATGACAGC-3′. RT-PCR was performed in triplicate wells from three independent biological experiments. Expression levels were normalized to β-actin and fold change was determined by comparative $C_T$ method.

### Image acquisition
Images were captured on a Nikon C2 confocal microscope with an electrically heated chamber for the Nikon motorized stage. Stage top incubator's controller for application with premixed gas. It includes a thermal controller, a humidity module, and a single-channel gas flow rate regulator. It is outfitted with brightfield and phase contrast with a Kinetix CMOS camera and fluorescent channels DAPI, GFP, and Texas Red. The Nikon LUN4 has a four-line solid-state laser system with Perfect Focus and DU3 High Sensitivity Detector System. The CFI60 Apochromat Lambda S 40× water immersion objective lens, N.A. 1.25, W.D. 0.16–0.2 mm, F.O.V. 22 mm, DIC, correction collar 0.15–0.19 mm, spring-loaded was used for confocal image acquisition. The CFI Plan Fluor DLL 40× N.A. 0.75, W.D. 0.66 mm was used for time-lapse imaging. Nikon NIS-Element Package and ImageJ were used for image analysis.

### Statistical analysis
For comparisons between two groups, a two-tailed, unpaired Student's $t$ test was used to determine statistical significance. Ordinary one-way ANOVA followed by Tukey's multiple comparisons were used for three or more groups using GraphPad Prism software. Data distribution was assumed to be normal, but this was not formally tested. Additional details are available in the figure legends. Differences in means were considered significant if $P < 0.05$ and designated as the following $P < 0.05$—*; $P < 0.01$—**; $P < 0.001$—***. $P < 0.0001$—****; ns—not significant.

### Online supplemental material
Fig. S1 shows that NAP1 KD for 36 h in DLD-1 cells halts cell division in prophase and prometaphase resulting in mitotic errors. Fig. S2 shows that mitosis does not elicit an innate immune response. Fig. S3 shows principal component analysis (PCA) of the phosphoproteome data and experimentally derived substrate sequence specificity for the Aurora family, PAK famliy, and TBK1 and IKKε kinases. Fig. S4 shows that phosphoproteomic data uncovers downstream substrates from other mitotic kinases. Fig. S5 shows gene ontology terms enrichment analysis and associated enrichment map networks for enriched phosphosites for centrosomal and microtubule-associated proteins. Video 1 shows

an untreated FKBP$^{F36V}$-NAP1 cell dividing. Videos 2, 3, 4, and 5 show dTAG$^v$-1 treated FKBP$^{F36V}$-NAP1 cells dividing with various mitotic and cytokinetic defects. Data S1 is an Excel file containing fractionated phospho-enriched samples and TMTpro analysis of WT or TBK1 KO lysates treated or not with MRT67397 compound. Data S2 is an Excel file containing the fractionated samples and TMTpro analysis of WT or TBK1 KO lysates treated or not with MRT67397 compound. Data S3 is an Excel file containing the phosphorylation sites analyzed for kinase prediction. Data S4 is an Excel file with the prediction match for the AZI2 S318 phosphorylation site to the 303 human kinome experimentally derived substrate motifs.

### Data availability

The MS TMT proteomic supporting data have been deposited to the MassIVE repository with the dataset identifier MSV000093220. Constructs used for this study will be available through https://Addgene.org. All other reagents, data, and material requests will be fulfilled by the corresponding author, A.M. Pickrell, PhD. The data underlying all figures and tables are available in the published article and its online supplemental material. This study did not generate code.

## Acknowledgments

We thank Dr. Daniela Cimini and Mathew Bloomfield for their advice, reagents, and expertise.

This work was supported by the National Institutes of Health Grants GM142368 (A.M. Pickrell) and departmental startup funds (A.M. Pickrell).

Author contributions: S. Paul performed planning and methodology, experimentation, data analysis, writing of the original draft, and review and editing. S.A. Sarraf performed planning and methodology, experimentation, data analysis, provided reagents, and review and editing. K.H. Nam performed experiments and data analysis. L. Zavar, N. DeFoor, S.R. Biswas, and L.E. Fritsch performed experiments. T.M. Yaron, J.L. Johnson, and E.M. Huntsman performed data analysis. A. Ordureau performed planning and methodology, experimentation, data analysis, and review and editing. L.C. Cantley provided computational software for analysis. A.M. Pickrell conceptualized the project, performed planning and methodology, experimentation, data analysis, writing of the original draft, and review and editing. All authors have read and approved the manuscript.

Disclosures: All authors have completed and submitted the ICMJE Form for Disclosure of Potential Conflicts of Interest. T.M. Yaron reported "other" from DeStroke, Inc. outside the submitted work. J.L. Johnson reported personal fees from Scorpion Therapeutics and personal fees from Volastra Therapeutics outside the submitted work. L.C. Cantley reported consulting fees from Scorpion Therapeutics, Volastra Therapeutics, and Larkspur. No other disclosures were reported.

Submitted: 20 March 2023

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

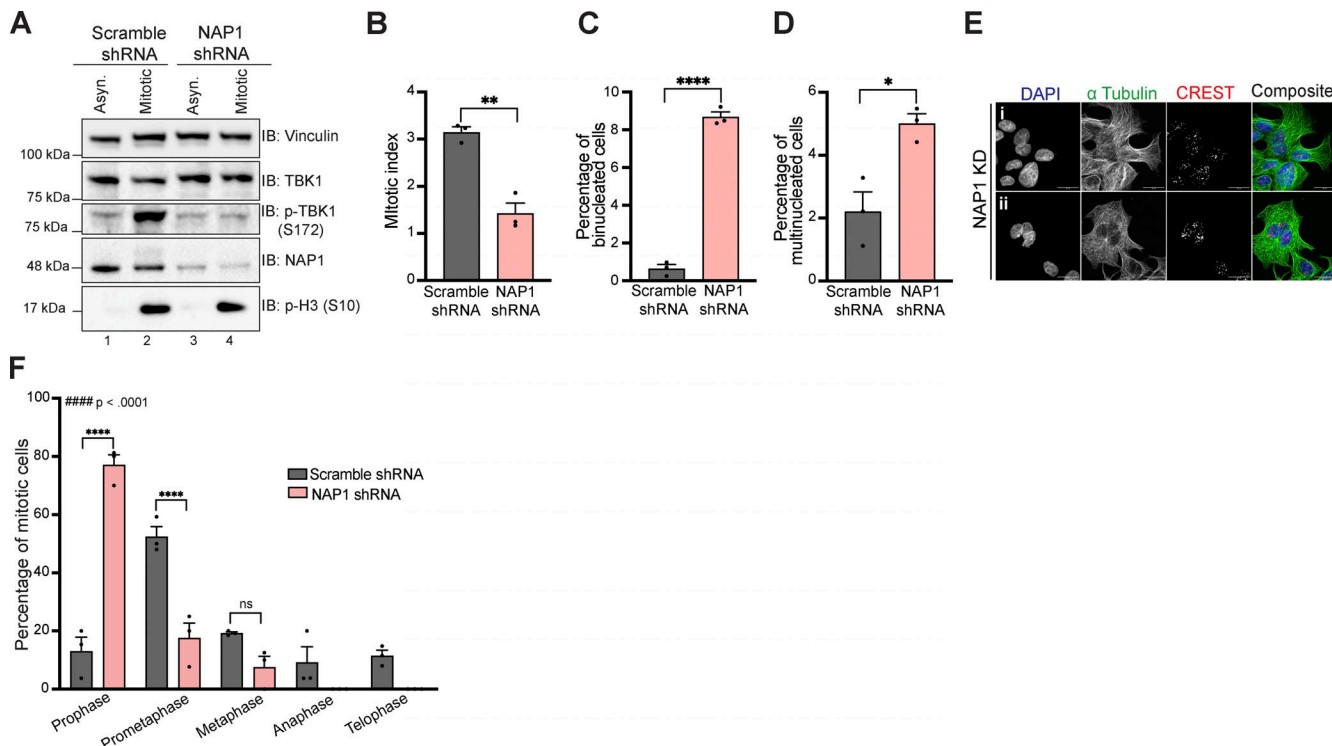

Figure S1. **NAP1 KD for 36 h in DLD-1 cells halts cell division in prophase and prometaphase resulting in mitotic errors. (A)** Western blot analysis of p-TBK1 levels in asynchronous and mitotic cells from scramble and NAP1 shRNA KD in DLD-1 cells over 36 h. RO-3306 was used for synchronization. **(B–D)** Percentage of mitotic (B), binucleated (C), and multinucleated (D) cells from an asynchronous population of either scramble or NAP1 KD for 36 h in DLD-1 cells. Error bars indicate ±SEM; $n = 3$ independent experiments. Random fields of view were captured sampling ~800 cells per biological replicate from each category. **(E)** Representative confocal images of NAP1 KD cells: (i) binucleated and (ii) multinucleated cells. DAPI (blue) was used as a nuclear counterstain, $a$-tubulin for cytoskeleton staining (green), and CREST for kinetochore staining (red). Scale bar, 20 µm. **(F)** Mitotic stage frequency distribution for scramble or NAP1 KD for 36 h in DLD-1 cells. Error bars indicate ±SEM; $n = 3$ independent experiments. Random fields of view were captured sampling ~800 cells per biological replicate from each category. Unpaired Student's $t$ test was performed for all statistical analysis (B–D and F). For F, Student's $t$ test compared the difference between groups during each phase of mitosis. * $P < .05$, ** $P < .01$, **** $P < .0001$, ns = not significant. Kolmogorov-Smirnov nonparametric test was used to analyze the differences between the frequency distribution between scramble and NAP1 KD in F. #### $p < .0001$. Source data are available for this figure: SourceData FS1.

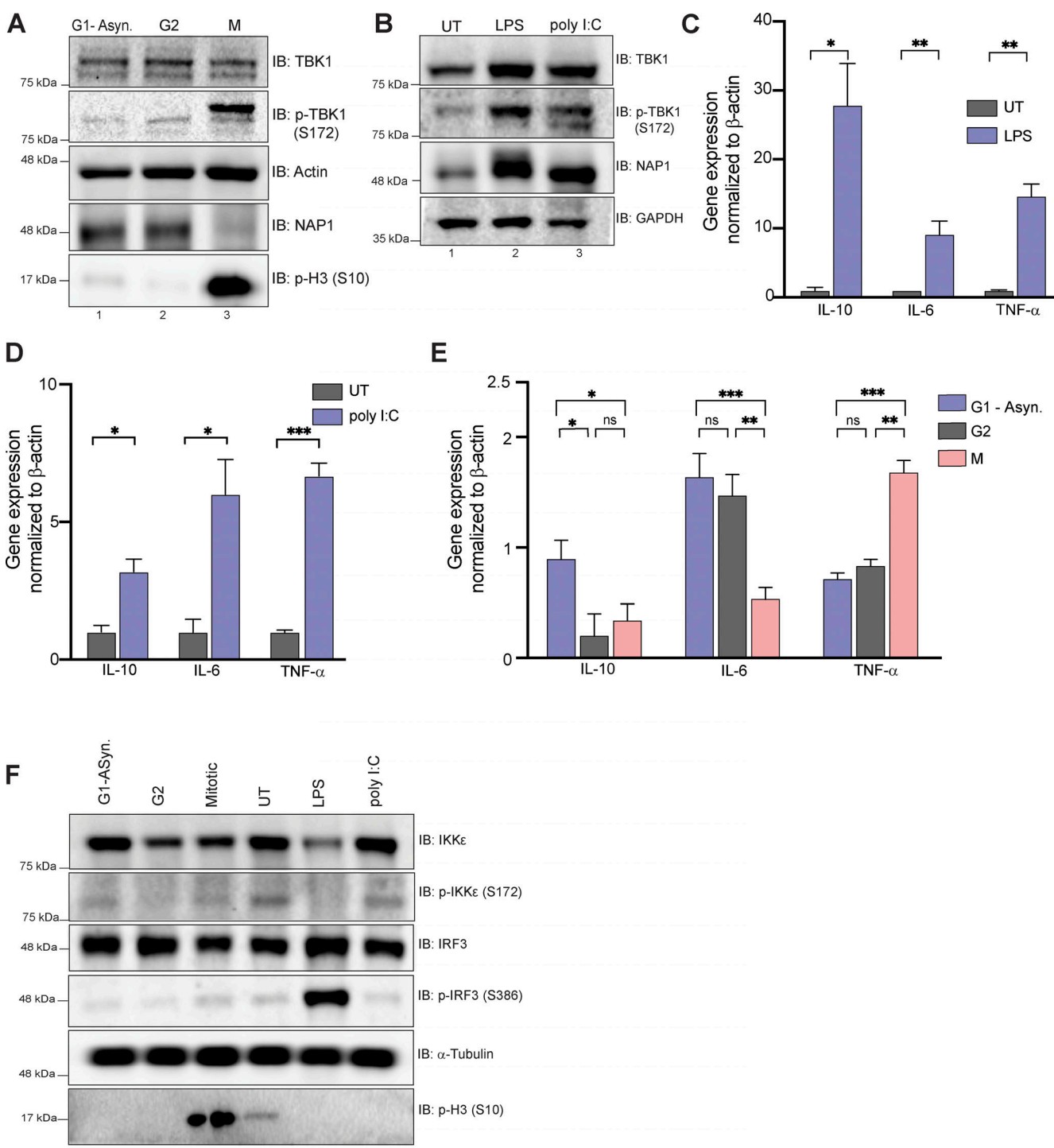

Figure S2. **Mitosis does not elicit an innate immune response. (A)** Western blot analysis of THP-1 cells synchronized at G2, M, or G1-asynchronous to determine p-TBK1 levels. Cells were synchronized at G2 using RO-3306. **(B)** Western blot analysis of THP-1 cells stimulated with LPS or poly I:C for 1 h and 8 h, respectively. Blots were probed for p-TBK1 and NAP1. **(C–D)** Relative mRNA expression of cytokines normalized to β-actin in THP-1 cells when stimulated with LPS (C) for 1 h or poly I:C (D) for 8 h. Error bars indicate ±SD; $n$ = 3 independent experiments. **(E)** Relative mRNA expression of cytokines during different cell cycle stages normalized to β-actin in THP-1 cells. Error bars indicate ±SD; $n$ = 3 independent experiments. Cells were synchronized at G2 using RO-3306. **(F)** Western blot analysis of p-IRF3 and p-IKKe in THP-1 cells synchronized at G2, M, or G1-asynchronous or stimulated with LPS or poly I:C for 1 h and 8 h, respectively. Cells were synchronized at G2 using RO-3306. Student's $t$ test (C and D) or one way ANOVA (E) was performed for all statistical analysis. * P < .05, ** P < .01, *** P < .001, ns = not significant. Source data are available for this figure: SourceData FS2.

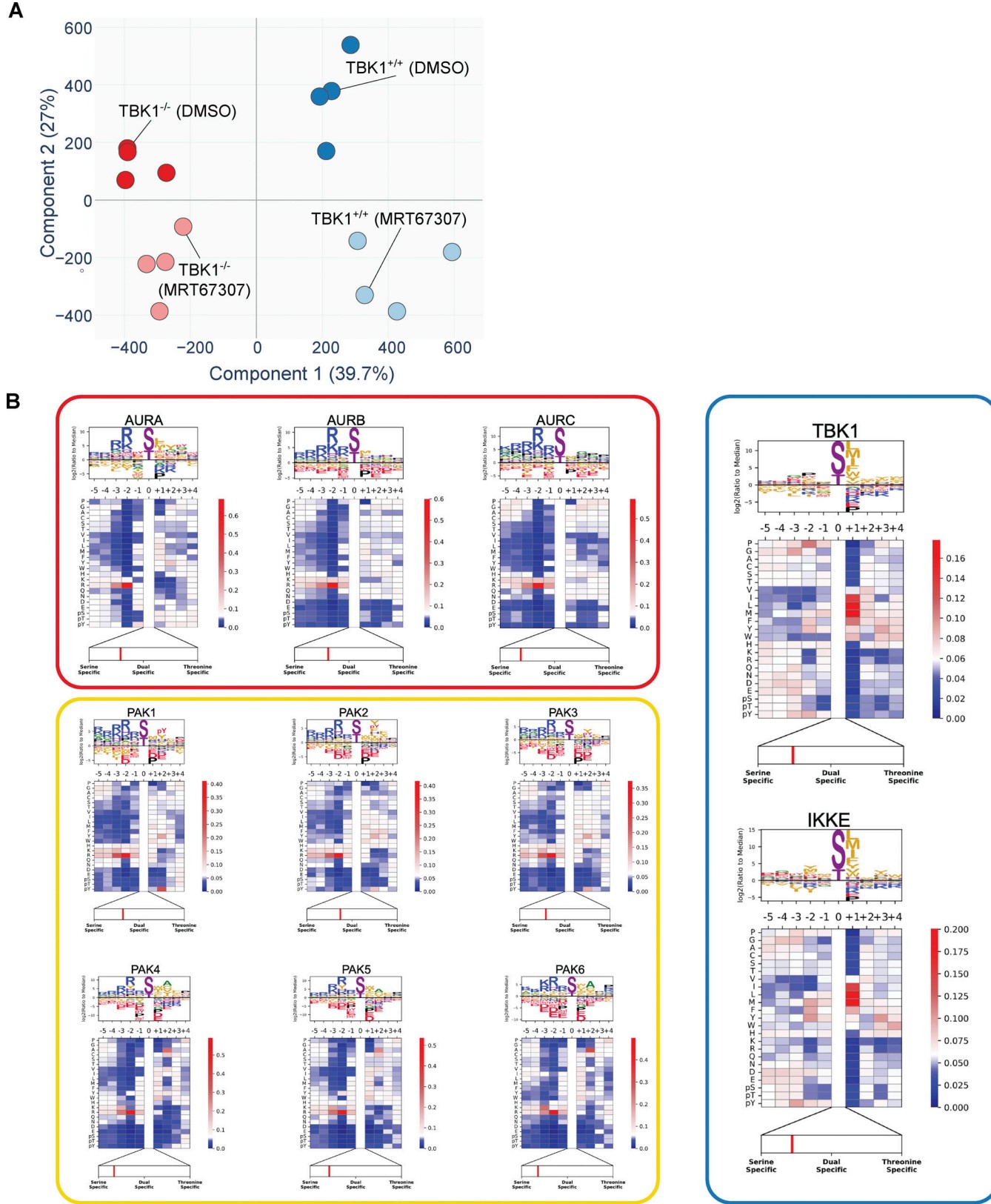

Figure S3. **Principal component analysis (PCA) of the phosphoproteome data and experimentally derived substrate sequence specificity for Aurora family, PAK famliy, TBK1, and IKKε kinases. (A)** Principal component analysis (PCA) of the phosphoproteome data. Replicate samples are shown separately. 39.7% of the changes in phosphoabundance are provided by Component 1, which represents the genetic background component, while 27% of the change are provided by Component 2, which represents the small molecule inhibitortreatment. **(B)** Individual motifs for experimentally derived substrate sequence specificity for Aurora family, PAK family, TBK1 and IKKε kinases. Data derived from Johnson et al. (2023).

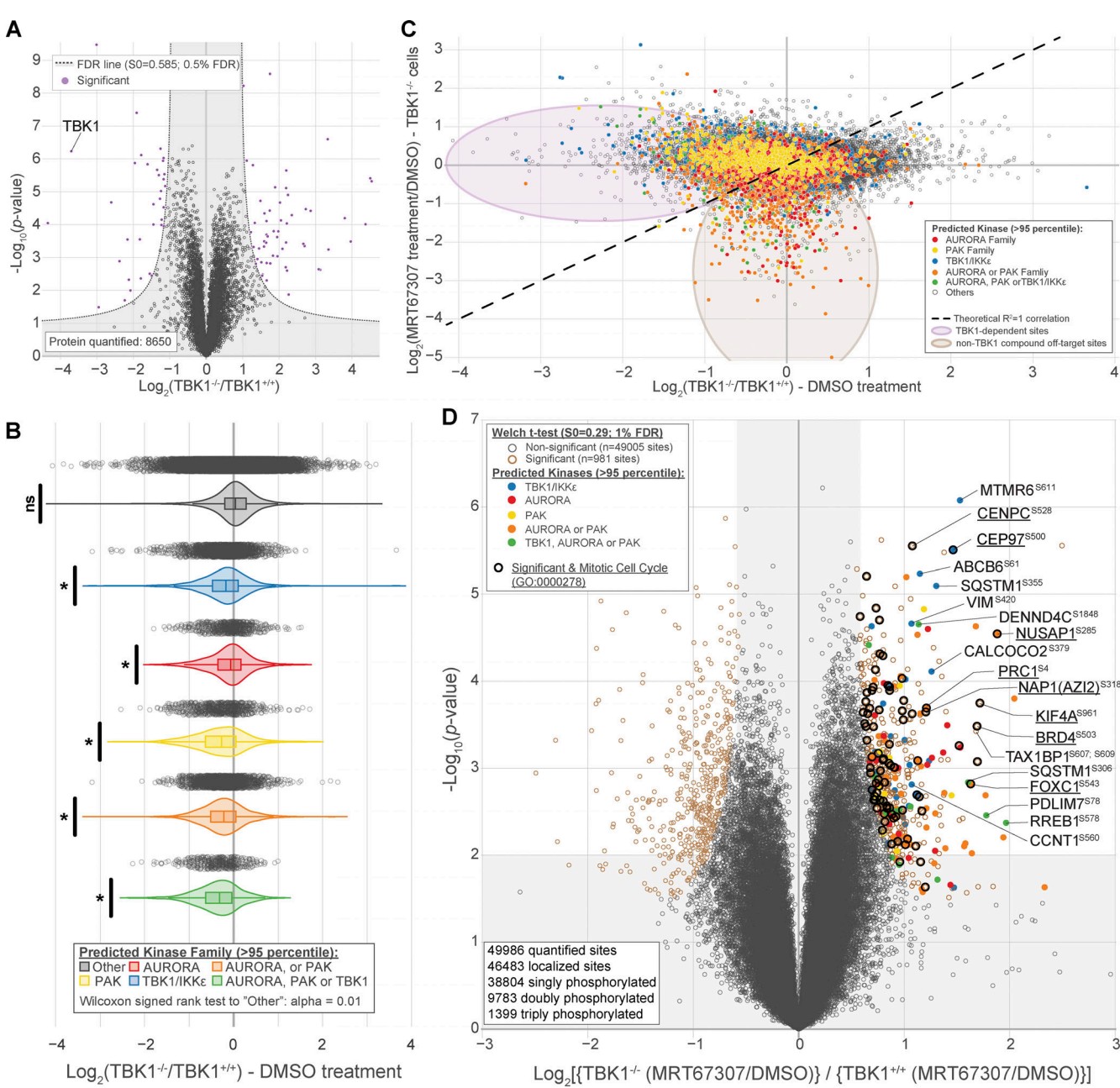

Figure S4. **Phosphoproteomic data uncovers downstream substrates from other mitotic kinases. (A)** Volcano plots [Log10 (P value) versus Log2 ratio] representing protein abundance that are affected by the loss of TBK1. Proteins are shown in black circles. The inset indicated additional color coding for the statistical analysis. **(B)** Violin plot of identified phosphorylation substrates grouped by predicted kinase family representing phosphorylation sites affected by the loss of TBK1. Proteins are shown in gray open circles. Wilcoxon signed rank test to "Other": alpha = 0.01. * P < 0.5, ns = not significant. **(C)** Volcano plots [Log2 ratio versus Log2 ratio] for representing phosphorylation sites that are affected by MRT67307 in TBK1 KO cells to distinguish impacts solely due to pharmacological treatment or genotype. Proteins are shown in gray open circles. Circles were color coded for the motif that most likely fit TBK1 (blue), Aurora kinases (red), PAK kinases (yellow), Aurora or PAK kinases (orange), or all three (green). **(D)** Volcano plots [Log10 (P value) versus Log2 ratio] for representing phosphorylation sites that are affected byMRT67307 and loss of TBK1. Proteins are shown in black circles (non-significant) and red circle (Tier 1 significant). Circles were color coded for the motif that most likely fit TBK1 (blue), Aurora kinases (red), PAK kinases (yellow), Aurora or PAK kinases (orange), or all three (green). Bolded black circles categorize known mitotic proteins (GO:0000278). The inset indicated additional color coding for the statistical analysis.

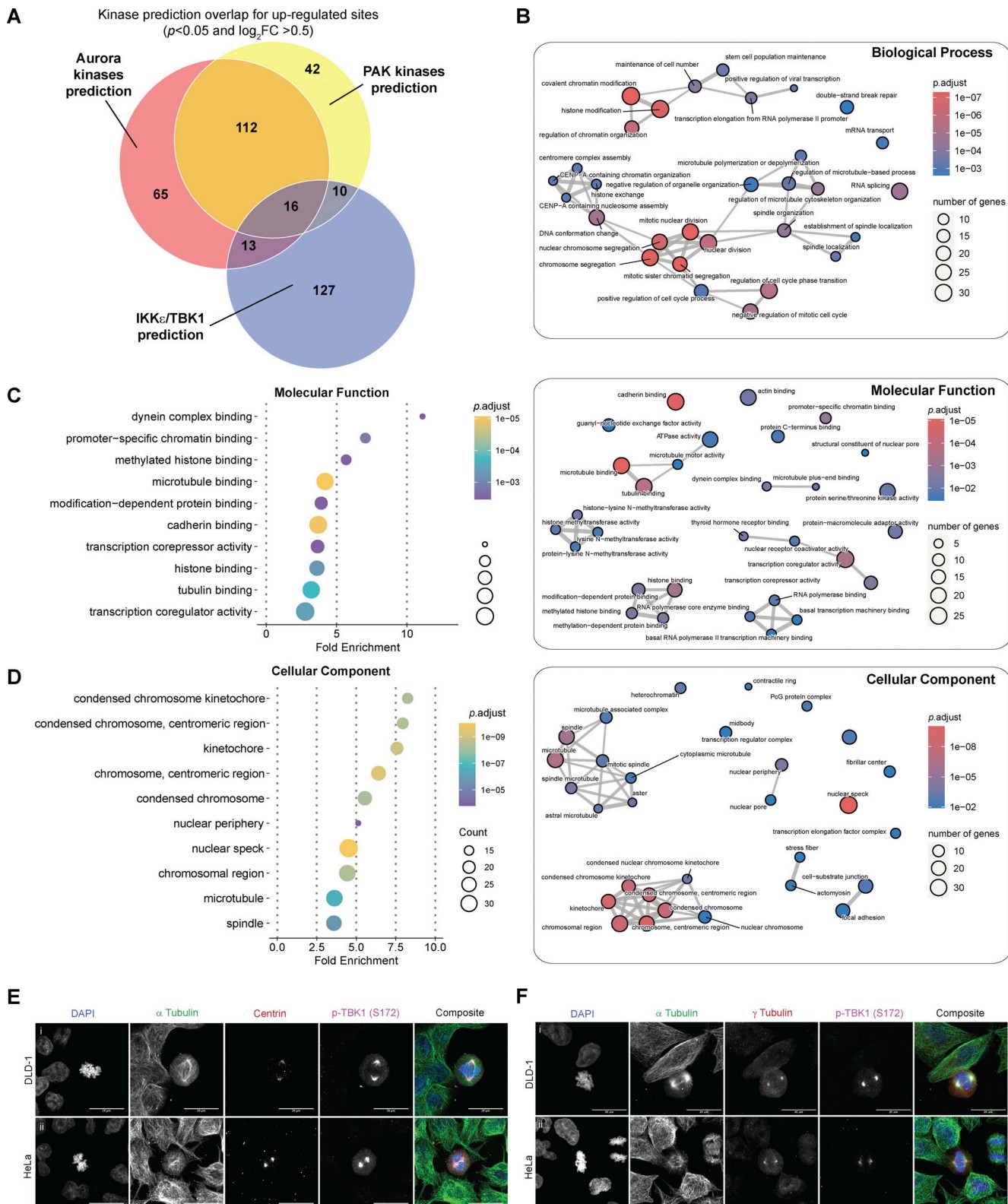

Figure S5. **Gene Ontology terms enrichment analysis and associated enrichment map networks for enriched phosphosites for centrosomal and microtubule associated proteins. (A)** Venn diagram illustrating the kinase prediction overlap for upregulated sites passing the P value cutoff of P < 0.05 and log2 ratio cutoff of +0.5. **(B–D)** Gene Ontology terms enrichment analysis and associated enrichment map networks for enriched phosphor-sites (Tier 2). For enrichment map networks, each node represents a gene set (i.e., a GO term) and each edge represents the overlap between two gene sets. **(E)** Representative confocal images of centrin (red) and p-TBK1 (magenta) staining in (i) DLD-1 and (ii) HeLa cells. DAPI (blue) was used as a nuclear counterstain and a-tubulin for cytoskeleton staining (green). Scale bar, 20 µm. **(F)** Representative confocal images of g-tubulin (red) and p-TBK1 (magenta) staining in (i) DLD-1 and (ii) HeLa cells. DAPI (blue) was used as a nuclear counterstain and a-tubulin for cytoskeleton staining (green). Scale bar, 20 µm.

Video 1.  **Untreated FKBP^F36V^-NAP1 cell dividing.** 4 frames per second.

Video 2.  **dTAG^v^-1-treated FKBP^F36V^-NAP1 cells dividing with various mitotic and cytokinetic defects.** 4 frames per second.

Video 3.  **dTAG^v^-1-treated FKBP^F36V^-NAP1 cells dividing with various mitotic and cytokinetic defects.** 4 frames per second.

Video 4.  **dTAG^v^-1-treated FKBP^F36V^-NAP1 cells dividing with various mitotic and cytokinetic defects.** 4 frames per second.

Video 5.  **dTAG^v^-1-treated FKBP^F36V^-NAP1 cells dividing with various mitotic and cytokinetic defects.** 4 frames per second.

**Provided online are Data S1, Data S2, Data S3, and Data S4. Data S1 is an Excel file containing fractionated phospho-enriched samples and TMTpro analysis of WT or TBK1 KO lysates treated or not with MRT67397 compound. Data S2 is an Excel file containing the fractionated samples and TMTpro analysis of WT or TBK1 KO lysates treated or not with the MRT67397 compound. Data S3 is an Excel file containing the phosphorylation sites analyzed for kinase prediction. Data S4 is an Excel file with the prediction match for the AZI2 S318 phosphorylation site to the 303 human kinome experimentally derived substrate motifs.**

