## [Peer Review File · The Journal of Cell Biology]

NAK-associated protein 1/NAP1 activates TBK1 to ensure accurate mitosis and cytokinesis

Swagatika Paul, Shireen Sarraf, Ki Hong Nam, Leila Zavar, Nicole DeFoor, Sahitya Ranjan Biswas, Lauren Fritsch, Tomer Yaron, Jared Johnson, Emily Huntsman, Lewis Cantley, Alban Ordureau, and Alicia Pickrell

Corresponding Author(s): Alicia Pickrell, Virginia Tech

Review Timeline:

Submission Date:	2023-03-20
Editorial Decision:	2023-04-19
Revision Received:	2023-10-03
Editorial Decision:	2023-10-30
Revision Received:	2023-11-08

Monitoring Editor: Arshad Desai

Scientific Editor: Andrea Marat

Transaction Report:

DOI: <https://doi.org/10.1083/jcb.202303082>

April 19, 2023

Re: JCB manuscript #202303082

Dr. Alicia Pickrell
Virginia Tech
Life Science I Room 217
970 Washington Street SW
Blacksburg, VA 24061

Dear Dr. Pickrell,

Thank you for submitting your manuscript entitled "NAK associated protein 1/NAP1 is required for mitosis and cytokinesis by activating TBK1". The manuscript has been evaluated by expert reviewers, whose reports are appended below. Unfortunately, after an assessment of the reviewer feedback, our editorial decision is against publication in JCB.

You will see that the reviewers all find the description of a role for NAP1 in regulating TBK1 activation during mitosis of potential interest for JCB. However, they have provided significant constructive feedback, addressing which seems essential to support the major conclusion and elevate the impact of the work. Notably, we agree that addressing their concerns regarding presentation as well as removing data reflecting potential off-target effects is necessary, as is using the dTAG system for depletion in a time-controlled manner. Altogether, while the manuscript presents an intriguing set of observations, we feel that addressing the numerous points raised by the reviewers reflects a more substantial effort than can be addressed in a typical revision period. If you wish to expedite publication of the current data, it may be best to pursue publication at another journal.

Given interest in the topic, we would be open to resubmission to JCB of a significantly revised and extended manuscript that fully addresses the reviewers' concerns and is subjected to further peer-review. If you would like to resubmit this work to JCB, please contact the journal office to discuss an appeal of this decision or you may submit an appeal directly through our manuscript submission system. Please note that priority and novelty would be reassessed at resubmission.

Regardless of how you choose to proceed, we hope that the comments below will prove constructive as your work progresses. We would be happy to discuss the reviewer comments further once you've had a chance to consider the points raised in this letter. You can contact the journal office with any questions, cellbio@rockefeller.edu or call (212) 327-8588.

Thank you for thinking of JCB as an appropriate place to publish your work.

Sincerely,

Arshad Desai, PhD
Monitoring Editor

Andrea L. Marat, PhD
Senior Scientific Editor

Journal of Cell Biology

Reviewer #1 (Comments to the Authors (Required)):

In this study, Swagatika et al. describe the analysis of TBK1 activation by the protein NAP1, and propose a function in mitosis. The first section of their manuscript addresses which of the many TBK1 interaction partners contributes to its activation in mitosis. This part includes some interesting data showing NAP1 localises to centrosomes and plays a role in regulating TBK1 activity, which is followed using a pTBK1-S172 antibody to the activation segment of the kinase. Removal of NAP1 using shRNA or a dTAG degenon results in reduced TBK1 autophosphorylation on S172, and mitotic spindle and cytokinesis defects in up to 10% of cells. The second section of the manuscript uses proteomics to identify TBK1 substrates in mitosis, before shifting back to the analysis of NAP1 phosphorylation by TBK1. The final conclusion is that TBK1 controls NAP1 stability by phosphorylation at S318.

Overall, the work has a lot of potential, but is confusing as presented and the cell biological analysis of the function of NAP1-TBK1 in mitosis is preliminary. I am also concerned that the inhibitor used for proteomics has a direct effect on Aurora kinase activity (see comment 7).

1. I would advise the authors to combine Figures 1, 2 and 3A-3F showing the effects of different TBK1 partners on its activation in mitosis, with a description of the main cell biological effects on mitosis in a new Figure 2. A lot of data is redundant and thus the main message gets a bit lost.
2. In Figure 1 it would be useful to show graphs for the effects of TANK, SINTBAD and NAP1 shRNA on TBK1 and p-TBK1 (S172) (Figure 1D-1F). That would better support the conclusions drawn from this figure about TBK1 activation in mitosis. SINTBAD appears to have an effect in asynchronous cells but not mitosis (Figure 1E).
3. Figure 2G and 2H are effectively the same thing and could be combined into one graph for bi/multinucleate cells (see comment 1). From the images shown in Figure 2J there wasn't an obvious increase in nuclear defects and this point needs better support, and time lapse imaging would be useful. Selected abnormal cells in mitosis are highlighted in Figure 2J and it would be helpful if the percentage of each phenotype was noted next to the image panels. One general comment, it is important to show both TBK and pTBK localise to the same structures, and not only show pTBK1 S172. Is the pTBK1 S172 signal lost in the HeLa TBK1 KO cell line?
4. The use of a degron to destabilise NAP1 is a very nice approach, and Figure 3G-3U dTAG should be a figure in its own right. However, the cells should be stained for TBK1 and pTBK1-S172 - the latter should be lost once the degron is activated. The NAP1 shRNA and data from NAP1 KO cell lines should be either in the revised Figure 1 or Figure 2. Time lapse imaging of mitotic cells to show how the phenotype develops would be informative.
5. The tissue blots in Figure 4A don't add much to the cell biology shown in the rest of the paper and without further justification are not needed. The other data in Figure 4 looks at the role of the NAP1-TBK1 interaction in TBK1 activity. This analysis isn't complete - the control in Figure 4M should be a stable cell line expressing GFP-NAP1 not untransfected cells. The cell biological part should show a side-by-side comparison of untreated mitotic cells, mitotic cells depleted of NAP1, and NAP1 depleted cells expressing GFP-NAP1 and GFP-NAP1 D230-270 with staining for TBK1 and pTBK1 S172. The message from the supporting Figure S5 is not clear to me. The title says it indicates pTBK1 is not only at centrosomes, using HeLa and DLD1 cells. The only markers used is gamma-tubulin which seems to overlap well with pTBK1 S172 in all cases. The graphs are for experiments shown in Figure 4K-4O, and use area rather than intensity (compare to Figure 4O).
6. Figure 5 is not convincing if the aim is to demonstrate cell cycle regulation of NAP1 expression. NAP1 appears to be present in both async and mitotic samples, possibly slightly reduced in mitosis. That may be due to increased turnover in mitosis rather than altered expression.
7. Figure 6 shows a proteomic analysis to identify TBK1 substrates in mitosis. Based on this data and some motif analysis the authors conclude that TBK1 has an effect on Aurora A and Aurora B kinase activities. Neither kinase is shown in the volcano plot, which is an important point for that conclusion. Figure 7A, S4E/F show that the TBK1 inhibitor MRT67307 results in complete loss of histone H3 S10 phosphorylation in different cell lines. However, TBK1 knockout cell lines have normal histone H3 S10 phosphorylation (Figure S4G). Thus, it is likely that an off-target effect of MRT67307 on Aurora kinases explains the proteomics data showing enrichment for Aurora sites. Only NAP1 phosphorylation is followed up, so this data could actually be removed.
8. Figure 7 shows analysis of NAP1 phosphorylation at S318, but no functional cell biology showing the impact on mitotic and cytokinesis. This should be extended to more fully the idea that centrosome localised TBK1 is necessary for mitosis and cytokinesis.

Reviewer #2 (Comments to the Authors (Required)):

In this study, Paul and colleagues identified NAP1 as an adaptor protein required for TBK1 activation. Through a series of experiments, they show that loss of either NAP1 or TBK1 results in the accumulation of binucleated and multinucleated cells, due to the several mitotic and cytokinetic defects that they were able to observe across several cell lines. Importantly, they also found that NAP1 levels during mitosis are tightly regulated by TBK1, through phosphorylation of NAP1 on serine 318 which in turn triggers proteasomal degradation (UPS). Finally, through unbiased quantitative phosphoproteomics analysis during mitosis, they uncovered unidentified TBK1 substrates, which implicate its upstream effects on other cell cycle kinases such as Aurora A and Aurora B. Overall, this is an interesting study and might be of interest for the readership of JCB, after properly addressing several major and minor points that I have listed below:

Major points

1. Quantifications of western blots are missing. The authors should state the number of replicates executed for each experiment.
2. Figure 1F: it looks like the mitotic fraction of sample #4 is lower than the one of sample #2. It is important to have similar amount of mitotic cells to do a proper comparison between samples. The decrease in pTBK1 might well be a consequence of cell cycle stage (mitotic vs. interphase) rather than driven by NAP1 depletion. This is even more pronounced in Figure 2A (compare sample #2 with #4 and #6) where the authors employed KO clones. Careful comparison should be made by harvesting mitotic cells (please see point 3 below). Further, on top of p-H3S10, the authors might consider using another mitotic marker, such as CyclinB1.
3. It is not clear how mitotic cells were collected. From the methods section, the authors state: "Nocodazole treatment: Cells were incubated with 1ug/ml nocodazole (Sigma) containing medium to synchronize at the G2/M border for 16 hours and collected for further experiments". It would be important to know whether a mitotic shake-off was performed or the entire population (mitotic and interphase cells) was harvested.

4. Other spindle drugs (taxol or monastrol) should be used to enrich for mitotic cells.
5. The authors show that NAP1 KO leads to slower growth (Figure 2E-I). In which cell cycle stage are those cells arrested/slowed down?
6. It would be important to check whether and how unperturbed mitotic progression (in terms of timing) is altered following NAP1 KO/KD. A live cell imaging experiment would address this. If mitotic length is increased in NAP1 KO/KD, it would be interesting to check whether there is an activation of SAC.

Minor points

1. Line numbers might be helpful
2. In the first paragraph of results (line number 9) "mitosis" is mis-spelled.
3. When DLD-1 cell line is introduced for the first time (page 6), it sounds like they are untransformed ("We attempted to use the non-transformed near diploid RPE-1 and DLD-1 cell lines"). Rephrasing this sentence might be helpful to clarify that DLD-1 are cancer cells, although near diploid.
4. From the methods section, the authors state: "Nocodazole treatment: Cells were incubated with 1ug/ml nocodazole (Sigma) containing medium to synchronize at the G2/M border for 16 hours and collected for further experiments". Please note that Nocodazole arrests cells in prometaphase, thus it's a mitotic arrest rather than a G2/M. Please change the text accordingly.

Reviewer #3 (Comments to the Authors (Required)):

The manuscript by Paul et al, demonstrated compelling data that NAK Associated Protein I (NAP1) is an important regulator of Tank Binding Kinase I (TBK1) in mitosis. TBK1 has several adaptor proteins that regulate its activity. The authors show that sustained loss of NAP1 leads to loss of TBK1 activity and subsequent mitotic defects. The authors also strongly demonstrate that TBK1 and NAP1 interact in mitosis. This was to me one of the most interesting and convincing aspects of the paper. The authors also use quantitative phospho-proteomics to identify TBK1 substrates in mitosis. These data strongly demonstrate an important function for TBK1 in mitosis in regulating Aurora B. Last, the authors suggest that NAP1 levels are regulated by TBK1, because TBK1 activity promotes degradation of NAP1 in mitosis.

Although this manuscript presents interesting and important points, there are several underlying stories, confusing, and overreaching statements that overinterpret the results and it is lacking critical controls at times which makes this reviewer much less enthusiastic about the manuscript.

I have major concerns with some of the data and conclusions that the authors make, as detailed below.

1. Although authors convincingly demonstrate that NAP1 is important in mitosis for maintain TBK1 activity, the authors do not demonstrate that this is mitotic specific. ShRNA, KO and even dTAG degradation of NAP1 was done over both interphase and mitosis. It is possible, that NAP1 activity from interphase continues through into mitosis and is important for sustaining TBK1 activity. The possibility and limitation of this study because of this of this is should be explicitly addressed or the direct degradation of NAP1 in mitosis using a system such as a dTAG needs to be done to demonstrate that this is mitotic specific.
2. The authors use p-H3 S10, a substrate of Aurora B as a marker for mitosis. While in some blots, including 1F, have robust signal, although somewhat reduced (as expected). It makes it a difficult marker to use for mitosis. The authors should consider using other markers of mitosis for their experiments.
3. On the same point, it is unclear why p-H3 S10 is variable within many of these western blots. For example, there is a mild reduction in Fig. 1F after NAP1 shRNA, and again in Fig. 2A and 2D, but then there is no visible change in p-H3 S10 in Fig. 3A and 3H. This is inconsistent with the conclusion that TBK1 promotes Aurora B activity and should be carefully addressed.
4. Moreover, the NAP1 rescue depicted in Fig. 2D demonstrates that TBK1 activity may be restored per p-TBK1 but the p-H3 S10 data would suggest that there is either insufficient activation or other mechanisms that preclude Aurora B activity restoration. It is also possible that the cells treated in S4E and S4F are no longer in mitosis, as p-H3 is responsive to Aurora B phosphorylation. Thus, the authors should carefully uncouple Aurora B activity and presence in mitosis in all western blots.
5. It is unclear how the authors quantified mitotic defects. The mitotic defects analyzed are in distinct stages of mitosis (i.e. monopolar spindle and acentric fragments) and should be analyzed separately. So, spindle polarity should be analyzed as a specific subset, chromosome mis-segregation events and types should be compared to each other, and not all clumped into one. This is especially true as it is difficult to know what normal spindle means, is that normal chromosome segregation, or bipolar spindle or both? Additionally, to analyze so many defects, it is not sufficient in this reviewer's opinion to analyze only 50 mitotic cells.
6. Acentric fragments suggest errors in replication and/or DNA damage in interphase which are unresolved, and cells enter mitosis, leaving the possibility that the authors are seeing effects that were lingering from interphase. This can be resolved if the authors deplete NAP1 in mitosis specifically.
7. Fig 5: The authors use CDK1i inhibitor synchronization into mitosis to determine the NAP1 levels in mitosis while coupling this with p-PLk1 T210 and p-CDK1 Y15. The authors should firstly quantify these changes in NAP1 levels as it is unclear by how much they are decreasing. Secondly, the authors are assuming using imperfect markers of mitosis, especially as it is unclear what percentage of cells were in mitosis after synchronization and how well the synchronization worked. To circumvent this the authors should couple their data with live-cell imaging or fixed cell analysis of the populations that they are sampling for their western blot. This is even more important in TAK243 treated cells p-CDK1 Y15 levels are not as high to begin with and the cells

have decreased p-CDK1 Y15 levels, thus it is possible that the effects from the TAK243 are due to perturbations in mitosis and not to UPS activity.

Minor points:

1. The authors should be clear in their methods about what they are considering spindle defects, defining acentric fragments, splayed spindles, ect.
2. There are several experiments that have only two replicates, a third biological replicate should be added.
3. There is not enough data to support the statement that NAP1 "has critical function like in cell division". The only thing that can be taken from these data are that NAP1 is not very abundantly or at all expressed in the small intestine and kidney.
4. Fig 4 O, there are two biological replicates, but it is unclear how the authors were able to achieve consistent average intensities between replicates. The authors should consider normalizing each replicate to the untreated control.
5. Fig S5: The authors should compare NAP1 KO to NAP1 Rescue, not both to WT to more clearly understand if NAP1 rescue construct does indeed affect p-TBK1.
6. The figure legend for S2 states that mitotic TBK does not elicit an immune response, however there are

On behalf of all the authors, we would like to thank the editors, editorial team, and the reviewers for spending their time and effort to read and prepare valuable comments for our manuscript. Our data describes a novel role for the innate immune response protein, NAK associated protein 1 (NAP1, also known as AZI2), during mitosis and cytokinesis. After screening the known adaptors for TBK1 from various other cellular processes, we provide key results demonstrating NAP1 binds to Tank Binding Kinase 1 (TBK1) to activate it during mitosis. We also provide evidence across multiple different cell lines using genetic and inducible manipulations to show that NAP1 localizes to the centrosomes during mitosis and regulates TBK1 for precise progression through cell division. To our knowledge, our report is the first to describe a non-canonical role of NAP1 in cell cycle regulation. After establishing NAP1 as key driver for mitotic progression, our additional work found that NAP1 levels are strictly regulated during mitosis, like other cell cycle regulatory proteins, via the ubiquitin proteasome system (UPS).

Upon revision, there were 3 major experimental revision requirements to address many of the reviewers' concerns. 1) Time lapse imaging is necessary for dTAG degraon NAP1 DLD-1 cells (Figure 3R-X, Videos 1-5). 2) Reanalysis of phosphoproteomic data comparing the TBK1 KO and parental cell line (Figure 3B, S4) 3) Repetition of certain experiments with quantification and additional controls (Figure 1D-L, 2A-E, 3B-D, 3F, 6F-I). The outcome of these experiments is outlined in more detail below and have only strengthened the support of our findings. We have attached a copy of the manuscript with revisions in blue text.

Additionally, our quantitative phosphoproteomics data provides a broad understanding of the complete landscape of proteins regulated by TBK1 during mitosis. This phosphoproteomic data in combination with analysis using data generated from *in vitro* peptide library motifs indicate that the influence that TBK1 has on mitosis and cytokinesis is partially due to its regulation of other major cell cycle regulatory proteins. Reviewer 1 had concerns about this data, which we acknowledge and have now addressed upon revision with additional analyses. Our paper has major implications not only for those studying cell division and cell cycle regulation, but due to NAP1 and TBK1's distinct subcellular location dynamics between mitosis, innate immune response, and selective autophagy pathways, we envision this paper will be valuable for those studying interorganelle communication, cancer, and immunology.

Reviewer #1 (Comments to the Authors (Required)):

In this study, Swagatika et al. describe the analysis of TBK1 activation by the protein NAP1, and propose a function in mitosis. The first section of their manuscript addresses which of the many TBK1 interaction partners contributes to its activation in mitosis. This part includes some interesting data showing NAP1 localises to centrosomes and plays a role in regulating TBK1 activity, which is followed using a pTBK1-S172 antibody to the activation segment of the kinase. Removal of NAP1 using shRNA or a dTAG degraon results in reduced TBK1 autophosphorylation on S172, and mitotic spindle and cytokinesis defects in up to 10% of cells. The second section of the manuscript uses proteomics to identify TBK1 substrates in mitosis, before shifting back to the analysis of

NAP1 phosphorylation by TBK1. The final conclusion is that TBK1 controls NAP1 stability by phosphorylation at S318.

Overall, the work has a lot of potential, but is confusing as presented and the cell biological analysis of the function of NAP1-TBK1 in mitosis is preliminary. I am also concerned that the inhibitor used for proteomics has a direct effect on Aurora kinase activity (see comment 7).

We thank the reviewer for their insightful comments and appreciation for the potential of our study. We have answered all the questions below to overcome any concerns about the manuscript. We have performed additional experiments that we felt strengthened the manuscript and concerns that this reviewer has about the data and its presentation.

1. I would advise the authors to combine Figures 1, 2 and 3A-3F showing the effects of different TBK1 partners on its activation in mitosis, with a description of the main cell biological effects on mitosis in a new Figure 2. A lot of data is redundant and thus the main message gets a bit lost.

We appreciate the reviewer's comment on the reorganization of the figures to show the effects of various TBK1 partners on activation during mitosis. Due to the limit on supplementary figures for JCB (which is 5), we kept Figure 1 as is, but we separated out Figure 2 with only the main biological effects on mitosis and cytokinesis in the HeLa cell knockouts combining data from Figure 2 and the previous Supplementary Figure 1. We then moved the DLD-1 NAP1 KD figure that was originally in Figure 3A-F to Supplementary Figure 1. Figure 3 now only shows data for the dTAG degron NAP1 DLD-1 cell line. We think this is the best compromise to better present the data as the Reviewer 1 suggested and to stay in the confines of the supplementary data requirement while adding the new revision experiments to the manuscript.

2. In Figure 1 it would be useful to show graphs for the effects of TANK, SINTBAD and NAP1 shRNA on TBK1 and p-TBK1 (S172) (Figure 1D-1F). That would better support the conclusions drawn from this figure about TBK1 activation in mitosis. SINTBAD appears to have an effect in asynchronous cells but not mitosis (Figure 1E).

We agree with the reviewer that a quantification of the effects of TANK, SINTBAD and NAP1 shRNA on TBK1 activation would better support the conclusions. We have now added the quantification graphs for these western blots with the change of p-TBK1 (S172) normalized to TBK1 levels taking into account the new mitosis loading control (see below) with 2 additional experimental replicates for each experiment. This data is now in Figure 1.

Upon additional replicates, we did see that SINTBAD shRNA did affect asynchronous cells (Figure 1F, H), so we also quantified these western blots taking into account the asynchronous signal normalized to the scramble control. We have mentioned this finding in the discussion on line 509.

3. Figure 2G and 2H are effectively the same thing and could be combined into one graph for bi/multinucleate cells (see comment 1). From the images shown in Figure 2J there wasn't an obvious increase in nuclear defects and this point needs better support, and time lapse imaging would be useful. Selected abnormal cells in mitosis are highlighted in Figure 2J and it would be helpful if the percentage of each phenotype was noted next to the image panels. One general comment, it is important to show both TBK and pTBK localise to the same structures, and not only show pTBK1 S172. Is the pTBK1 S172 signal lost in the HeLa TBK1 KO cell line?

We wanted to specify the distinction between binucleated and multinucleated cells to underscore the fact that cytokinetic failure often leads to binucleation, whereas multinucleation can be an indication of multiple mitotic/cytokinetic failure events over time. This distinction becomes relevant when we show the data from the 20 hours of NAP1 depletion in the dTAG NAP1 cell lines. There is a significant increase in the percentage of binucleated cells in the NAP1 depleted cells whereas percentage of multinucleated cells is indifferent as NAP1 loss was only evaluated over 1 round of cell division. We wish to keep this data in its current form of representation as separate graphs.

We agree with the reviewer that to gain insight into the defects caused by NAP1 depletion it is useful to have time lapse imaging. Thus, we performed a time lapse imaging with the dTAG NAP1 DLD-1 cell lines to better characterize mitotic progression after NAP1 depletion upon revision in Figure 3R-X, Videos 1-5).

The representative images from Figure 2J show an average of 10 cells; however, we have counted at least 1000 cells per biological replicate for mitotic and nuclear defects analysis, but to better visualize these defects we have added additional insets in Figure 2J, updated the Material and Methods, and added these percentages to the insets on Figure 2 and 3 as well as these percentages being on the pie charts.

The reviewer has pointed out the TBK1 signal should colocalize with p-TBK1 (S172) in the cells. However, this is not a trivial point. It is still unclear if all available TBK1 molecules colocalizes at centrosomes during mitosis, or if it is just locally activated. Previous publications [1, 2] examine TBK1 in the contexts of mitophagy have overexpressed tagged TBK1 to determine the localization of inactivated TBK1, but in our experience, this causes abnormal activation [3]. TBK1 antibodies do not produce reliable immunocytochemistry staining (Figure 1, in this letter). We were excited to see a recently published paper in Cell Reports [4] that used methanol/acetone fixation to detect TBK1 with a new commercial Abcam TBK1 antibody. However, in our hands, we did not see a difference in the staining pattern between WT and KO cells (Figure 1, in this letter). Future studies will need to generate an endogenously tagged TBK1 cell line to address where inactivated TBK1 is during mitosis or if all TBK1 are activated at the centrosome during mitosis. We have included IF staining showing p-TBK1 (S172) staining in TBK1 rescue, TBK1 KO and TBK1 S172A rescue cells (Figure 2, in this letter).

Figure 1:

Figure 1: TBK1 immunostaining in WT HeLa and TBK1 KO cells. DAPI (blue) was used for nuclear staining with alpha tubulin for cytoskeleton (red) and TBK1 (green). Scale bar = 20uM).

Figure 2:

Figure 2: p-TBK1 (S172) immunostaining in WT TBK1 rescue, TBK1 KO and S172A TBK1 rescue lines. DAPI (blue) was used for nuclear staining and p-TBK1 (S172) (red) was used for activated TBK1. Mitotic cells are highlighted with *; scale bar = 20uM).

4. The use of a degron to destabilise NAP1 is a very nice approach, and Figure 3G-3U dTAG should be a figure in its own right. However, the cells should be stained for TBK1 and pTBK1-S172 - the latter should be lost once the degron is activated. The NAP1 shRNA and data from NAP1 KO cell lines should be either in the revised Figure 1 or Figure 2. Time lapse imaging of mitotic cells to show how the phenotype develops would be informative.

We agree with the reviewer and have addressed these comments above about figure organization and time lapse imaging. We have also performed IF for p-TBK1 with dTAG treatment and upon a 4-hour washout which this data is in Figure 4O-S.

5. The tissue blots in Figure 4A don't add much to the cell biology shown in the rest of the paper and without further justification are not needed. The other data in Figure 4 looks at the role of the NAP1-TBK1 interaction in TBK1 activity. This analysis isn't complete - the control in Figure 4M should be a stable cell line expressing GFP-NAP1 not untransfected cells. The cell biological part should show a side-by-side comparison of untreated mitotic cells, mitotic cells depleted of NAP1, and NAP1 depleted cells expressing GFP-NAP1 and GFP-NAP1 D230-270 with staining for TBK1 and pTBK1 S172. The message from the supporting Figure S5 is not clear to me. The title says it indicates pTBK1 is not only at centrosomes, using HeLa and DLD1 cells. The only markers used is gamma-tubulin which seems to overlap well with pTBK1 S172 in all cases. The graphs are for experiments shown in Figure 4K-4O, and use area rather than intensity (compare to Figure 4O).

We added the human tissue blot since NAP1 protein expression levels have not been reported in the literature to our knowledge, but considering that Reviewer 1 and Reviewer 3 were in agreement that the data did not add to the manuscript, we removed it upon revision.

We originally did not analyze Figure 4M as suggested because the intensity of p-TBK1 in NAP1 full length rescue and \$\Delta\$ NAP1 rescue varied depend on the amount of NAP1 expressed; however, this was another concern of Reviewer 3 (see below). We attempted to make a rescue line upon revision with an EGFP tagged full length NAP1, but expression did not match evenly even with sorting strategies and single colony selection. Considering this, we decided to remove this data and the discussion of this data. Future studies will utilize knockin mutant lines and time lapse imaging to better determine p-TBK1 and TBK1 kinetics to the centrosome.

We apologize for the confusion. We also used centrin in Figure S5 to demonstrate that the activated TBK1 area was outside the boundary of the centrosomes. We rewrote this

portion of the results on lines 388-392 to make it clear as to the points we are trying to convey.

6. Figure 5 is not convincing if the aim is to demonstrate cell cycle regulation of NAP1 expression. NAP1 appears to be present in both async and mitotic samples, possibly slightly reduced in mitosis. That may be due to increased turnover in mitosis rather than altered expression.

We respectively do not completely understand this comment. We have demonstrated that NAP1 is relatively stable, transcription is mostly repressed during mitosis [5], and autophagy, which can be repressed during mitosis [6], is not responsible for NAP1 degradation (Figure 5). Our data suggests that a complete loss of NAP1 is not viable in diploid cell lines (see Results and Discussion). We believe the treatment with MG132 in Figure 5 demonstrates UPS turnovers NAP1 during mitosis dependent on S318 phosphorylation. We changed the title of the figure legend to “Fig. 5. NAP1 expression level is controlled by the UPS” to better explain this point.

7. Figure 6 shows a proteomic analysis to identify TBK1 substrates in mitosis. Based on this data and some motif analysis the authors conclude that TBK1 has an effect on Aurora A and Aurora B kinase activities. Neither kinase is shown in the volcano plot, which is an important point for that conclusion. Figure 7A, S4E/F show that the TBK1 inhibitor MRT67307 results in complete loss of histone H3 S10 phosphorylation in different cell lines. However, TBK1 knockout cell lines have normal histone H3 S10 phosphorylation (Figure S4G). Thus, it is likely that an off-target effect of MRT67307 on Aurora kinases explains the proteomics data showing enrichment for Aurora sites. Only NAP1 phosphorylation is followed up, so this data could actually be removed.

To address these concerns from the reviewer, we have now added a volcano plot with just the significant downregulated proteins and sites in Figure 6B for the genotype comparison, added a new Figure S4 to demonstrate off-target effects due to inhibitor and genotype, and verified aurora kinase activity was downregulated in TBK1 KO and Aurora B activity was decreased in dTAG treated degenon degradable NAP1 DLD-1 cell line (Figure 6F-I). We have also added Figure S4B taking into account that all the substrates identified as TBK1, Aurora, or PAK substrates in this study are downregulated indicating that are affected upon the loss of TBK1. We do see a loss of phosphorylation of both Aurora A and B in our proteomic screen, but they do not reach significance (Data S1). Future work will be needed to understand how TBK1 is affecting Aurora and PAK kinases.

8. Figure 7 shows analysis of NAP1 phosphorylation at S318, but no functional cell biology showing the impact on mitotic and cytokinesis. This should be extended to more fully the idea that centrosome localised TBK1 is necessary for mitosis and cytokinesis.

We thank the reviewer for their suggestion, but we do not believe that we can fully rescue the NAP1 KO HeLa cell line. Although we recovered pTBK1 activity in Figure 2C-D, pH3 levels never recover (see comment from another reviewer below, original Figure 2D). We performed pTBK1 immunostaining in the S318 NAP1 cell line and saw some cells that

appeared normal, but other cells where pTBK1 staining patterns were not completely localized to the centrosomes (Figure 3, in this letter). Considering these issues, future additional studies with gene edited cell lines will be required to further understand the functional consequence of increased NAP1 stability and loss of S318 phosphorylation.

Figure 3:

Figure 3: p-TBK1 (S172) immunostaining in NAP1 and NAP1 S318A rescue lines. DAPI (blue) was used for nuclear staining, alpha-tubulin (red) for cytoskeleton, and p-TBK1 (S172) (green) was used for activated TBK1. Scale bar = 20uM, inset = 5uM.

Reviewer #2 (Comments to the Authors (Required)):

In this study, Paul and colleagues identified NAP1 as an adaptor protein required for TBK1 activation. Through a series of experiments, they show that loss of either NAP1 or TBK1 results in the accumulation of binucleated and multinucleated cells, due to the several mitotic and cytokinetic defects that they were able to observe across several cell lines. Importantly, they also found that NAP1 levels during mitosis are tightly regulated by TBK1, through phosphorylation of NAP1 on serine 318 which in turn triggers proteasomal degradation (UPS). Finally, through unbiased quantitative phosphoproteomics analysis during mitosis, they uncovered unidentified TBK1 substrates, which implicate its upstream effects on other cell cycle kinases such as Aurora A and Aurora B. Overall, this is an interesting study and might be of interest for

the readership of JCB, after properly addressing several major and minor points that I have listed below:

The authors appreciate the reviewer's comments and time taken to review our manuscript. We thank the reviewer for stating our work as an interesting study and worked to address all the concerns that the reviewer has pointed below with the additional experimentation.

Major points

1. Quantifications of western blots are missing. The authors should state the number of replicates executed for each experiment.

We added WB quantification graphs increasing the number of experimental replicates for the revised main figures as the reviewer suggested for any westerns not done so already. In addition to increasing the number experimental replicates for western blotting, we used another mitotic marker (see below). Some main figures we did not perform additional experimental replicates because we did these experiments in multiple cell lines (Figure 5E-M), or explained below why it was technically difficult but had the proper controls.

2. Figure 1F: it looks like the mitotic fraction of sample #4 is lower than the one of sample #2. It is important to have similar amount of mitotic cells to do a proper comparison between samples. The decrease in pTBK1 might well be a consequence of cell cycle stage (mitotic vs. interphase) rather than driven by NAP1 depletion. This is even more pronounced in Figure 2A (compare sample #2 with #4 and #6) where the authors employed KO clones. Careful comparison should be made by harvesting mitotic cells (please see point 3 below). Further, on top of p-H3S10, the authors might consider using another mitotic marker, such as CyclinB1.

To address the concerns of the reviewer about comparing the same number of mitotic cells, we used another mitotic marker and repeated many of the main figure western blots (see above). However, we believe that we did avoid collecting interphase cells (see below), but our data does suggest that with the knowledge we have now about TBK1 phosphorylation targets, p-H3 S10 was a poor choice, which we did not know at the time when performing all of the experiments when starting this project. As suggested by the reviewer, cyclin B1 was unchanged upon the loss of TBK1 and NAP1 as it wasn't a TBK1 or Aurora substrate, so we repeated experiments used cyclin B1 as our mitotic marker.

3. It is not clear how mitotic cells were collected. From the methods section, the authors state: "Nocodazole treatment: Cells were incubated with 1ug/ml nocodazole (Sigma) containing medium to synchronize at the G2/M border for 16 hours and collected for further experiments". It would be important to know whether a mitotic shake-off was performed or the entire population (mitotic and interphase cells) was harvested.

We thank the reviewer for their suggestion to elaborate on our methods for mitotic cell collection. In the case of both RO-3306 and nocodazole treatment, the cells were

collected with a mitotic shake. Remaining cells on the cell culture plate were discarded to eliminate the interphase cells from mitotic sample collection.

We elaborated on our method for mitotic cell collection in the methods section for better understanding of our experimental procedure.

4. Other spindle drugs (taxol or monastrol) should be used to enrich for mitotic cells.

We appreciate the reviewer's suggestion for using a different spindle drug to enrich mitotic cells; however, to overcome the criticism if our data is a result of the cytotoxic impact of spindle depolarizing drug nocodazole, we have also used the cytostatic drug RO-3306 to synchronize the cells at G2 and release the cells in normal growth medium to collect mitotic cells within one hour of G2 release using mitotic shake.

5. The authors show that NAP1 KO leads to slower growth (Figure 2E-I). In which cell cycle stage are those cells arrested/slowed down?

We have now performed time lapse imaging in the dTAG NAP1 cell line (see Material and Methods, Figure 3R-X, Videos 1-5). We found in the dTAG NAP1 cell line that treated cells spend a significantly higher amount of time in prophase and prometaphase. Although we didn't do the live imaging for the NAP1 KO cell line, our data in the 36-hour NAP1 KD is in agreement (Figure S1F).

6. It would be important to check whether and how unperturbed mitotic progression (in terms of timing) is altered following NAP1 KO/KD. A live cell imaging experiment would address this. If mitotic length is increased in NAP1 KO/KD, it would be interesting to check whether there is an activation of SAC.

From our live cell imaging we found that the overall mitotic length is increased after NAP1 depletion. However, the duration between metaphase plate formation and the onset of chromosome segregation did not appear significantly different between normal and NAP1 depleted cells (Figure 3R-V). Additionally, NAP1 KD DLD-1 cells did not have many cells make it into metaphase, so we did not check the activation of SAC.

Minor points

1. Line numbers might be helpful

Line numbers have been added to the manuscript.

2. In the first paragraph of results (line number 9) "mitosis" is mis-spelled.

Thank you, we have fixed this spelling in the revised manuscript.

3. When DLD-1 cell line is introduced for the first time (page 6), it sounds like they are untransformed ("We attempted to use the non-transformed near diploid RPE-1 and DLD-1 cell lines"). Rephrasing this sentence might be helpful to clarify that DLD-1 are cancer cells, although near diploid.

We rephrased this line to “We attempted to use the near diploid cells RPE-1 and DLD-1 cell lines”

4. From the methods section, the authors state: "Nocodazole treatment: Cells were incubated with 1ug/ml nocodazole (Sigma) containing medium to synchronize at the G2/M border for 16 hours and collected for further experiments". Please note that Nocodazole arrests cells in prometaphase, thus it's a mitotic arrest rather than a G2/M. Please change the text accordingly.

We thank the reviewer for pointing this out. We replaced G2/M with prometaphase.

Reviewer #3 (Comments to the Authors (Required)):

The manuscript by Paul et al, demonstrated compelling data that NAK Associated Protein I (NAP1) is an important regulator of Tank Binding Kinase I (TBK1) in mitosis. TBK1 has several adaptor proteins that regulate its activity. The authors show that sustained loss of NAP1 leads to loss of TBK1 activity and subsequent mitotic defects. The authors also strongly demonstrate that TBK1 and NAP1 interact in mitosis. This was to me one of the most interesting and convincing aspects of the paper. The authors also use quantitative phospho-proteomics to identify TBK1 substrates in mitosis. These data strongly demonstrate an important function for TBK1 in mitosis in regulating Aurora B. Last, the authors suggest that NAP1 levels are regulated by TBK1, because TBK1 activity promotes degradation of NAP1 in mitosis.

Although this manuscript presents interesting and important points, there are several underlying stories, confusing, and overreaching statements that overinterpret the results and it is lacking critical controls at times which makes this reviewer much less enthusiastic about the manuscript.

We thank the reviewer for their appreciation of our work that the reviewer found to be convincing and interesting. We hope we addressed the reviewer's comments and concerns by adding additional controls and additional phosphoproteomic analyses to improve the quality of our manuscript.

I have major concerns with some of the data and conclusions that the authors make, as detailed below.

1. Although authors convincingly demonstrate that NAP1 is important in mitosis for maintain TBK1 activity, the authors do not demonstrate that this is mitotic specific. ShRNA, KO and even dTAG degradation of NAP1 was done over both interphase and mitosis. It is possible, that NAP1 activity from interphase continues through into mitosis and is important for sustaining TBK1 activity. The possibility and limitation of this study because of this of this is should be explicitly addressed or the direct degradation of NAP1 in mitosis using a system such as a dTAG needs to be done to demonstrate that this is mitotic specific.

To overcome the limitation of depleted NAP1 function in interphase, we performed time lapse imaging with the dTAG NAP1 DLD-1 cell lines to capture mitotic progression after NAP1 depletion. Considering that dTAG takes 1 hour for degradation, we depleted NAP1 for 1 hr and imaged cells for 4-5 hours to capture mitotic cells post NAP1 depletion (Figure 3R-X, Videos 1-5). With this method we tried to eliminate or reduce the possibility of any interphase NAP1 depletion that could impact mitosis. This data nicely compliments our findings in the three other systems where we had to manipulate NAP1 in interphase.

2. The authors use p-H3 S10, a substrate of Aurora B as a marker for mitosis. While in some blots, including 1F, have robust signal, although somewhat reduced (as expected). It makes it a difficult marker to use for mitosis. The authors should consider using other markers of mitosis for their experiments.

To address the criticism over the usage of p-H3 S10 as a mitotic marker which we addressed above in our response to Reviewer 2, we used cyclin B1 for our main western blot figures that we had to repeat.

3. On the same point, it is unclear why p-H3 S10 is variable within many of these western blots. For example, there is a mild reduction in Fig. 1F after NAP1 shRNA, and again in Fig. 2A and 2D, but then there is no visible change in p-H3 S10 in Fig. 3A and 3H. This is inconsistent with the conclusion that TBK1 promotes Aurora B activity and should be carefully addressed.

We agree with the reviewers that this is an inconsistency. The phosphoproteomics and figures that the reviewer is referring to are in HeLa, but this inconsistency the reviewer is discussing are results from the DLD-1 cell line. To ensure that this was not an error on our part, we performed aurora kinase activity western blotting in the TBK1 KO HeLa and dTAG NAP1 DLD-1 cell line. We saw that Aurora A and B activity was significantly decreased in TBK1 KO cells, and Aurora B activity is disrupted in dTAG NAP1 DLD-1 cells when treated for 2 hours prior to and during release (Figure 6F-I). Future research directions will elucidate this connection between TBK1 and Aurora kinase activity and may reveal why there is this discrepancy between cell lines.

4. Moreover, the NAP1 rescue depicted in Fig. 2D demonstrates that TBK1 activity may be restored per p-TBK1 but the p-H3 S10 data would suggest that there is either insufficient activation or other mechanisms that preclude Aurora B activity restoration. It is also possible that the cells treated in S4E and S4F are no longer in mitosis, as pH3 is responsive to Aurora B phosphorylation. Thus, the authors should carefully uncouple Aurora B activity and presence in mitosis in all western blots.

We agree, and considering that the NAP1 KO accumulates many defects, we do not believe that just adding back NAP1 will restore all the problems this cell line accumulates over time. However, NAP1 levels are slightly elevated in the rescue even though we attempted to tier the virus to obtain rescue levels closest to the parental line. We believe now that the abolishment p-H3 S10 level in the original S4E and S4F figure is due to the MRT67307 drug (see above). We have removed these western blots, and have now

replaced them with dTAG treated NAP1 westerns blots demonstrating the loss of Aurora B activity (Figure 6H-I).

5. It is unclear how the authors quantified mitotic defects. The mitotic defects analyzed are in distinct stages of mitosis (i.e. monopolar spindle and acentric fragments) and should be analyzed separately. So, spindle polarity should be analyzed as a specific subset, chromosome mis-segregation events and types should be compared to each other, and not all clumped into one. This is especially true as it is difficult to know what normal spindle means, is that normal chromosome segregation, or bipolar spindle or both? Additionally, to analyze so many defects, it is not sufficient in this reviewer's opinion to analyze only 50 mitotic cells.

For our mitotic defects analysis, we have followed the guidelines outlined from [7] and have gotten advice from Dr. Daniela Cimini, an expert in mitosis and cell cycle, who we have acknowledged. In this study, we have considered normal spindle as bipolar spindle with the correct orientation in the prometaphase and metaphase cells. Any error in chromosome segregation with bipolar spindle has not been included under spindle defects. Therefore, all the names of all the mitotic defects have been mentioned with the mitotic stage of the cell (Figures 2K, 3M). To make our mitotic defects analysis clear, we have elaborated the methods of identifying each defect under the methods section (lines 670-683). We have grouped the spindle defects and chromosome segregation defects under the umbrella of mitotic defects to better represent all types of errors found in case of NAP1 loss of depletion.

We have counted 50 mitotic cells per biological replicate (n=3 independent experiments) per genotype (total 150 mitotic cells/ genotype).

6. Acentric fragments suggest errors in replication and/or DNA damage in interphase which are unresolved, and cells enter mitosis, leaving the possibility that the authors are seeing effects that were lingering from interphase. This can be resolved if the authors deplete NAP1 in mitosis specifically.

We initially did this to determine how depleted NAP1 over 1 division period was affected. However, we understand that this may cause unforeseen effects from interphase for mitotic and cytokinetic defects analysis, so we performed a time lapse imaging with the dTAG NAP1 DLD-1 cell lines and added dTAG 1hr before imaging (see above, Figure 3R-X, Videos 1-5). This was the shortest amount of time treated where NAP1 was reduced prior to release in our hands.

7. Fig 5: The authors use CDK1i inhibitor synchronization into mitosis to determine the NAP1 levels in mitosis while coupling this with p-PLk1 T210 and p-CDK1 Y15. The authors should firstly quantify these changes in NAP1 levels as it is unclear by how much they are decreasing. Secondly, the authors are assuming using imperfect markers of mitosis, especially as it is unclear what percentage of cells were in mitosis after synchronization and how well the synchronization worked. To circumvent this the authors should couple their data with live-cell imaging or fixed cell analysis of the populations that they are sampling for their western blot. This is even more important in

TAK243 treated cells p-CDK1 Y15 levels are not as high to begin with and the cells have decreased p-CDK1 Y15 levels, thus it is possible that the effects from the TAK243 are due to perturbations in mitosis and not to UPS activity.

We agree with the Reviewer that TAK243 will perturb mitosis, as all E1 activation will be perturbed, thus interfering with cyclin degradation, etc. However, we wanted an additional way besides using MG132 as in Figure 5 to show NAP1 degradation is UPS dependent. We removed the TAK243 portion of the blot. We performed this experiment to narrow down when NAP1 levels were decreasing. We had performed this blot multiple times, but the temporal aspect of this experiment made it difficult to release and collect; therefore, we showed the best representative western of this result. We will reiterate that this is only 1 experimental replication, but we do believe that this experiment provides important information as it corroborates with the new IF data in Figure 5O.

NAP1 endogenous staining is not possible from the antibodies that we have tried. However, considering that the NAP1 dTAG DLD-1 cell line is tagged, we performed HA staining and found 2 x HA tags are sufficient for IF detection and performed the experiment as suggested by the reviewer.

Minor points:

1. The authors should be clear in their methods about what they are considering spindle defects, defining acentric fragments, splayed spindles, ect.

We have updated the methods section, lines 680-683.

2. There are several experiments that have only two replicates, a third biological replicate should be added.

We added a third replicate for the many of the main western blot figures (see above) except for Figures 5E-N and 7C-F. We did not do this because these experiments were done three times in three different cell lines with the same result or with the phostag gels, these blots are technically difficult to get interpretable crisp bands, but did include a phosphatase control.

3. There is not enough data to support the statement that NAP1 "has critical function like in cell division". The only thing that can be taken from these data are that NAP1 is not very abundantly or at all expressed in the small intestine and kidney.

Reviewer 1 did not feel this data was necessary, so we removed it upon revision.

4. Fig 4 O, there are two biological replicates, but it is unclear how the authors were able to achieve consistent average intensities between replicates. The authors should consider normalizing each replicate to the untreated control.

All the steps involved in staining and capturing the confocal images were kept the same for both the biological replicates. We reanalyzed the data and normalized IF intensity and area in Figure 4M-N and R-S as suggested by the reviewer. We have removed this data

from the figure (see response to Reviewer 1) and have added a different dataset with the correct control which conveys the same information.

5. Fig S5: The authors should compare NAP1 KO to NAP1 rescue, not both to WT to more clearly understand if NAP1 rescue construct does indeed affect p-TBK1.

Reviewer 1 also had issue with this experiment. We were originally concerned that the rescue of NAP1 did not perfectly match the levels of the mutant. However, we omitted this data (see response to Reviewer 1). To demonstrate the rescue of pTBK1 activity, we used the dTAG NAP1 cell line for IF in Figure 4O-S.

6. The figure legend for S2 states that mitotic TBK does not elicit an immune response, however there are

This statement is cut off midway. We would like to see the complete sentence to respond accordingly. We checked on the online submission site, and also do not see the rest of this comment.

References

1. Richter, B., et al., *Phosphorylation of OPTN by TBK1 enhances its binding to Ub chains and promotes selective autophagy of damaged mitochondria*. Proc Natl Acad Sci U S A, 2016. **113**(15): p. 4039-44.
2. Moore, A.S. and E.L. Holzbaur, *Dynamic recruitment and activation of ALS-associated TBK1 with its target optineurin are required for efficient mitophagy*. Proc Natl Acad Sci U S A, 2016. **113**(24): p. E3349-58.
3. Sarraf, S.A., et al., *PINK1/Parkin Influences Cell Cycle by Sequestering TBK1 at Damaged Mitochondria, Inhibiting Mitosis*. Cell Rep, 2019. **29**(1): p. 225-235 e5.
4. Scrima, N., et al., *Rabies virus P protein binds to TBK1 and interferes with the formation of innate immunity-related liquid condensates*. Cell Rep, 2023. **42**(1): p. 111949.
5. Palozola, K.C., et al., *Mitotic transcription and waves of gene reactivation during mitotic exit*. Science, 2017. **358**(6359): p. 119-122.
6. Odle, R.I., et al., *An mTORC1-to-CDK1 Switch Maintains Autophagy Suppression during Mitosis*. Mol Cell, 2020. **77**(2): p. 228-240 e7.
7. Baudoin, N.C. and D. Cimini, *A guide to classifying mitotic stages and mitotic defects in fixed cells*. Chromosoma, 2018. **127**(2): p. 215-227.

October 30, 2023

RE: JCB Manuscript #202303082R-A

Dr. Alicia Pickrell
Virginia Tech
Life Science I Room 217
970 Washington Street SW
Blacksburg, VA 24061

Dear Dr. Pickrell:

Thank you for submitting your revised manuscript entitled "NAK-associated protein 1/NAP1 activates TBK1 to ensure accurate mitosis and cytokinesis". We would be happy to publish your paper in JCB pending final revisions necessary to meet our formatting guidelines (see details below).

When preparing the final version of the manuscript, we encourage you to consider the feedback from Reviewer 1 which highlights a weakness of the study that may detract from the strengths. We will let you decide on the best course of action with respect to this feedback but agree with the reviewer that potentially developing this aspect further and featuring it in a future study, rather than here, may represent a good course of action.

A. MANUSCRIPT ORGANIZATION AND FORMATTING:

Full guidelines are available on our Instructions for Authors page, <https://jcb.rupress.org/submission-guidelines#revised>.
Submission of a paper that does not conform to JCB guidelines will delay the acceptance of your manuscript.

- 1) Text limits: Character count for Articles is < 40,000, not including spaces. Count includes abstract, introduction, results, discussion, and acknowledgments. Count does not include title page, figure legends, materials and methods, references, tables, or supplemental legends.
- 2) Figures limits: Articles may have up to 10 main text figures.
- 3) Figure formatting: Scale bars must be present on all microscopy images, including inset magnifications. Molecular weight or nucleic acid size markers must be included on all gel electrophoresis.
- 4) Statistical analysis: Error bars on graphic representations of numerical data must be clearly described in the figure legend. The number of independent data points (n) represented in a graph must be indicated in the legend. Statistical methods should be explained in full in the materials and methods. For figures presenting pooled data the statistical measure should be defined in the figure legends. Please also be sure to indicate the statistical tests used in each of your experiments (either in the figure legend itself or in a separate methods section) as well as the parameters of the test (for example, if you ran a t-test, please indicate if it was one- or two-sided, etc.). Also, if you used parametric tests, please indicate if the data distribution was tested for normality (and if so, how). If not, you must state something to the effect that "Data distribution was assumed to be normal but this was not formally tested."
- 5) Abstract and title: The abstract should be no longer than 160 words and should communicate the significance of the paper for a general audience. The title should be less than 100 characters including spaces. Make the title concise but accessible to a general readership.
- 6) Materials and methods: Should be comprehensive and not simply reference a previous publication for details on how an experiment was performed. Please provide full descriptions in the text for readers who may not have access to referenced manuscripts.
- 7) Please be sure to provide the sequences for all of your primers/oligos and RNAi constructs in the materials and methods. You must also indicate in the methods the source, species, and catalog numbers (where appropriate) for all of your antibodies. Please also indicate the acquisition and quantification methods for immunoblotting/western blots.
- 8) Microscope image acquisition: The following information must be provided about the acquisition and processing of images:
 - a. Make and model of microscope
 - b. Type, magnification, and numerical aperture of the objective lenses

- c. Temperature
- d. Imaging medium
- e. Fluorochromes
- f. Camera make and model
- g. Acquisition software
- h. Any software used for image processing subsequent to data acquisition. Please include details and types of operations involved (e.g., type of deconvolution, 3D reconstitutions, surface or volume rendering, gamma adjustments, etc.).

10) Supplemental materials: There are strict limits on the allowable amount of supplemental data. Articles may have up to 5 supplemental figures. Please also note that tables, like figures, should be provided as individual, editable files. A summary of all supplemental material should appear at the end of the Materials and methods section.

13) ORCID IDs: ORCID IDs are unique identifiers allowing researchers to create a record of their various scholarly contributions in a single place. Please note that ORCID IDs are now *required* for all authors. At resubmission of your final files, please be sure to provide your ORCID ID and those of all co-authors.

Please note that JCB now requires authors to submit Source Data used to generate figures containing gels and Western blots with all revised manuscripts. This Source Data consists of fully uncropped and unprocessed images for each gel/blot displayed in the main and supplemental figures. Since your paper includes cropped gel and/or blot images, please be sure to provide one Source Data file for each figure that contains gels and/or blots along with your revised manuscript files. File names for Source Data figures should be alphanumeric without any spaces or special characters (i.e., SourceDataF#, where F# refers to the associated main figure number or SourceDataFS# for those associated with Supplementary figures). The lanes of the gels/blots should be labeled as they are in the associated figure, the place where cropping was applied should be marked (with a box), and molecular weight/size standards should be labeled wherever possible.

Journal of Cell Biology now requires a data availability statement for all research article submissions. These statements will be published in the article directly above the Acknowledgments. The statement should address all data underlying the research presented in the manuscript. Please visit the JCB instructions for authors for guidelines and examples of statements at (<https://rupress.org/jcb/pages/editorial-policies#data-availability-statement>).

B. FINAL FILES:

****It is JCB policy that if requested, original data images must be made available to the editors. Failure to provide original images upon request will result in unavoidable delays in publication. Please ensure that you have access to all original data images prior to final submission.****

****The license to publish form must be signed before your manuscript can be sent to production. A link to the electronic license to publish form will be sent to the corresponding author only. Please take a moment to check your funder requirements before choosing the appropriate license.****

Thank you for this interesting contribution, we look forward to publishing your paper in Journal of Cell Biology.

Sincerely,

Arshad Desai, PhD
Monitoring Editor

Andrea L. Marat, PhD
Senior Scientific Editor

Journal of Cell Biology

Reviewer #1 (Comments to the Authors (Required)):

The authors have carried out extensive revisions of the work and have addressed many of my initial comments. Importantly, this strengthens the conclusions that TBK1 and NAP1 have some role in cell division, even if the exact function is not defined.

The major weakness remains the link to Aurora kinases. In my view, it is too early to draw any conclusions about links to Aurora kinases, and that would be best left for future studies where those questions can be explored in detail. As I originally wrote, most of what is in Figure 6 could be removed without weakening the core message of the study. I hope this comment is helpful to the authors.

Reviewer #2 (Comments to the Authors (Required)):

The authors satisfactorily addressed all concerns raised by this reviewer. I support publication in JCB.

Reviewer #3 (Comments to the Authors (Required)):

The authors have performed an extensive amount of work to address all comments raised and covered them satisfactorily. Therefore I recommend publication of the manuscript.